# Technical note: Effects of iron(II) on fluorescence properties of dissolved organic matter at circumneutral pH

Kun Jia[1,*], Cara C.M. Manning[1,†], Ashlee Jollymore[2,‡], and Roger D. Beckie[1]

[1]University of British Columbia, Department of Earth, Ocean and Atmospheric Sciences, Vancouver, BC, Canada
[2]University of British Columbia, Institute for Resources, Environment and Sustainability, Vancouver, BC, Canada
*Currently at: AECOM Canada
†Currently at: Plymouth Marine Laboratory
‡Currently at: Province of British Columbia, Ministry of Forest, Lands, Natural Resource Operations and Rural Development

*Correspondence to*: Kun Jia (kun418@gmail.com) and Roger Beckie (rbeckie@eoas.ubc.ca)

**Abstract.** Modern fluorescence spectroscopy methods, including excitation-emission matrix (EEMs) spectra parsed using parallel factor analysis (PARAFAC) statistical approaches, are widely used to characterize dissolved organic matter (DOM) pools. The effect of soluble reduced iron, Fe(II), on EEM spectra can be significant, but is difficult to quantitatively assign. In this study, we examine the effects of Fe(II) on the EEM spectra of groundwater samples from an anaerobic deltaic aquifer containing up to 300 mg/L Fe(II), located a few kilometers from the ocean, adjacent to the Fraser River in Richmond, British Columbia, Canada. We added varying quantities of Fe(II) into groundwater samples to evaluate Fe(II)-DOM interactions. Both the overall fluorescence intensity and the intensity of the primary peak, a humic-like substance at excitation/emission wavelengths 239/441-450 nm (Peak A), decreased by approximately 60% as Fe(II) concentration increased from 1 to 306 mg/L. Furthermore, the quenching effect was non-linear and proportionally stronger at Fe(II) concentrations below 100 mg/L. This non-linear relationship suggests a static quenching mechanism. In addition, DOM fluorescence indices are substantially influenced by the Fe(II) concentration. With increasing Fe(II), the fluorescence index (FI) shifts to higher values, the humidification index (HIX) shifts to lower values, and the freshness index (FrI) shifts to higher values. Nevertheless, the 13-component PARAFAC model showed that the component distribution was relatively insensitive to Fe(II) concentration, and thus, PARAFAC may be a reliable method for obtaining information about the DOM composition and its redox status in Fe(II)-rich waters. By characterizing the impacts of up to 300 mg/L Fe(II) on EEMs using groundwater from an aquifer which contains similar Fe(II) concentrations, we advance previous works which characterized impacts of lower Fe(II) concentrations (less than 2 mg/L) on EEMs.

## 1 Introduction

Fluorescence spectroscopy has been widely used to characterize the properties of dissolved organic matter as it is highly sensitive to the structures and functional chemistry of aquatic organic matter (Baker & Spencer, 2004; Fellman et al., 2010; Helms et al., 2008; Stedmon & Bro, 2008; Weishaar et al., 2003). In this method, light at a known wavelength (the excitation wavelength) is passed through a sample, and the absorbance at that frequency and fluorescence (emission) at other frequencies

is measured. Such spectra can be used to derive commonly utilized fluorescence indices that correlate to specific forms of organic matter (Aiken, 2014; Coble et al., 2014; Hudson et al., 2007; Murphy et al., 2013). These indices include the fluorescence index (FI), which is calculated as the ratio between the emission at 470 nm to that at 530 nm at an excitation wavelength of 370 nm and relates to the concentration of aromatic, microbially derived lignin-like organic matter (McKnight et al., 2001). An excitation-emission matrix (EEM) is prepared by systematically repeating the measurements at a range of different excitation and emission wavelengths. These measurements are highly sensitive to the structures and functional chemistry of aquatic organic matter, which determine the unique pattern of peaks present within the EEM spectra (Aiken, 2014; Coble, 1996; Coble et al., 2014; Fellman et al., 2010). Due to the complexity of the EEM spectra obtained from each unique sample, a number of statistical methods have been used to decompose EEM spectra and relate emission patterns to functional chemistry of organic matter within a sample. Parallel factor analysis (PARAFAC) is a commonly utilized statistical means of compartmentalizing EEM spectra into discrete peaks that may then be compared to broad organic matter classes (Bro, 1997; Chen et al., 2010; Jaffé et al., 2014; Murphy et al., 2013; Stedmon & Bro, 2008).

It is well accepted that dissolved organic matter (DOM) fluorescence is quenched or enhanced by interactions with metal ions, including Fe(III) (Ohno et al., 2007; Poulin et al., 2014; Pullin et al., 2007; Senesi, 1990; Shen et al., 2020), Fe(II) (Poulin et al., 2014), Al(III) (Ohno et al., 2007), Cu(II) (Senesi, 1990; Shen et al., 2020) and Hg(II) (Senesi, 1990). Fe(III) is recognized as an important source of interference for fluorescence measurements (Ohno et al., 2007; Pullin et al., 2007). Previous studies have reported a significant quenching effect caused by the binding of Fe(III) to organic ligands. A possible mechanism that may account for quenching is the formation of organometal complexes at the fluorescent sites (Rue & Bruland, 1995; Senesi, 1990). Such complexes can efficiently decrease the fluorescence intensity of the fluorophore. Furthermore, the degree of quenching varies among different organic-matter compounds, which increases the complexity and uncertainty in characterizing and predicting the iron-binding effect across a range of DOM types (Ohno et al., 2007). However, limited research has focused on the quenching effect of Fe(II) interference in anoxic groundwater, where reducing conditions are present. Poulin et al. (2014) first demonstrated that Fe(II) and DOM can form organometal complexes that decrease fluorescence intensity. Their experiments were only designed to characterize the Fe(II) quenching effect for surface water with moderately elevated DOM concentrations (2.3 to 5.0 mg/L) under low Fe (II) concentrations (0-1.5 mg/L). To our knowledge, the extent of fluorescence quenching in groundwater with higher Fe(II) concentrations is not known.

The fluorescence quenching effect in Fe(II)-rich groundwater is still poorly understood and warrants further investigation, given the prevalence of high DOM and Fe(II) in groundwater in deltaic sediments (Bolton & Beckie, 2011) and sites contaminated with organics, for example, from landfills or fuel spills (van Breukelen & Griffioen, 2004; Christensen et al., 2001; Heron et al., 1994). In most instances, these high Fe(II) groundwaters are found when the oxidation of organic matter is coupled to solid-phase Fe(III) reduction, dissolving Fe(II) into groundwater at circumneutral pH. For example, 1.5-10 mg/L

Fe(II) in groundwater is commonly observed in the organic-rich groundwaters of the Bengal Basin (Harvey et al., 2002), and up to 90 mg/L Fe(II) has been observed in landfill leachate in the Netherlands (van Breukelen & Griffioen, 2004).

The objective of this study was to assess the influence of high concentrations of Fe(II) on the fluorescence properties of DOM by titrating up to 300 mg/L (5.4 mM) Fe(II) into groundwater collected from a deltaic aquifer in Richmond, British Columbia, Canada. This groundwater is representative of groundwater found in diagenetically immature, organic-rich deltaic sediments, where Fe(II) concentrations can reach up to 300 mg/L (Bolton & Beckie, 2011; Jia, 2015). The biogeochemistry of groundwater at this site, and an analysis of the origin of the extraordinarily high Fe(II) concentrations, are described in Jia (2015).

We identified the degree of quenching by Fe(II) based on the excitation-emission matrix (EEMs) regions and peaks. In this study we fit EEM spectra to a previously derived 13-component PARAFAC model (Cory & McKnight, 2005); see section 3.3 for further details.

## 2 Deltaic groundwater stock solution

We collected representative deltaic groundwaters from what is known as the Kidd 2 site, located adjacent to the Fraser River a few kilometers upstream from its outlet to the ocean, near Vancouver, Canada (49°11'53.34"N, 123° 6'53.25"W), where a near-surface sandy aquifer is found between 5 m and 22 m below ground surface (Bolton & Beckie, 2011; Jia, 2015). In the anaerobic deltaic aquifer, Fe(II) is released to groundwater by oxidation of dissolved organic matter (DOM) in a process that is affected by the circulation of saline ocean water. At the site, denser, saline ocean water enters the aquifer in the hyporheic zone at the river bottom, flows inland along the base of the aquifer to a maximum distance of approximately 500 m inland where it overturns flows back towards the river under a regional hydraulic gradient from freshwater recharged inland (Neilson-Welch & Smith, 2001), forming a wedge of saline water in the aquifer (Figure 2). Along the flow path, the saline water mixes with fresh groundwater. The saline-freshwater mixture eventually discharges to the river at the top of the saline wedge. Two mixing zones can be identified along the saline wedge: at the bottom of the wedge, freshwater from the lower confining silt flows up into the overlying sandy aquifer as the saline water flows inland (the "lower mixing zone") and at the top of the wedge, terrestrial recharge from inland flows on top of the saline water as it flows back to the river (the "upper mixing zone"). High concentrations of Fe(II) are observed along the circulation flow path, especially in the upper mixing zone, where pore water Fe(II) concentrations peak above 300 mg/L (5.4 mM) (Figure 2) (Jia, 2015).

## 3 Methodology

### 3.1 Sample collection

For the measurements in this study, we collected a single stock solution of representative natural DOM-containing groundwater, to which we added (titrated) increasing concentrations of Fe(II). We selected groundwater from W3-14 at a depth

of 20.03 m as the DOM-containing stock solution as it had the lowest Fe(II) concentration (1.3 mg/L) at the Kidd 2 site (Jia, 2015), allowing us to explore a large range of Fe(II) concentrations. The multilevel sampling port consisted of a 0.635 cm inner diameter low-density polyethylene tube with a 5 cm fiberglass-mesh screen (Neilson-Welch & Smith, 2001). Three tubing volumes of groundwater were purged with a peristaltic pump while pH was monitored using an OAKTON™ pH/mV/°C meter in a sealed flow-through cell to prevent degassing. The pH and temperature stabilized at 7.44 and 11°C, respectively.

The groundwater was filtered through 0.45 μm cellulose filters, then stored in a 1 L amber glass bottle with a Teflon-lined plastic cap, without acidification. The bottle was filled with no headspace and duct tape was used to further seal the sample and minimize the oxidation of Fe(II). The collected 1 L stock solution was refrigerated at 4°C until fluorescence analysis (within ~14 days). Although some degradation of the DOM may have occurred during the holding period, this would not significantly affect our conclusions as our intention was to determine how Fe(II) addition affects the fluorescence properties of DOM, rather than to characterize the properties of DOM at the Kidd 2 site. The stock solution DOM concentration of 10.7 mg/L was measured using high temperature combustion with a HACH™ IL 550 TOC-TN analyzer, detection limit 1 mg/L, at the Environmental Engineering Laboratory in UBC's Department of Civil Engineering.

### 3.2 Fe(II) addition experiment and concentration determination

Experimental solutions were prepared from the stock DOM solution in an anaerobic glove box (Coy Labs, MI, USA), filled with an $N_2/H_2$ mixture (95% of $N_2$ and 5% of $H_2$) with a palladium catalyst inside the chamber, which maintains gaseous $O_2$ levels of less than 5 ppm. The stock DOM solution was deaerated in the glove box by purging with pure $N_2$ for 30 minutes. The sample addition experiments used glass cuvettes that were acid-washed with 10% $HNO_3$, and rinsed with deionized distilled water.

As the highest observed Fe(II) concentration in the groundwater at the Kidd 2 site was approximately 300 mg/L (Jia, 2015), the Fe(II) addition experiment was designed for a range of Fe(II) concentrations (from 1 to 300 mg/L). An Fe spiking solution of 1000 mg/L Fe(II) was prepared with $FeSO_4(H_2O)_7$, following Poulin et al. (2014), using the DOM stock solution so that spiking with Fe(II) would not change the overall concentration of DOM. Experiments were performed by sequentially adding Fe(II) spiking solution to an initial volume of 250 mL of DOM stock solution to reach 10 different concentrations between 1.3 to 306 mg/L. Previous analyses of water from W3-14 via ICP-OES (Jia, 2015) indicated the $SO_4^{2-}$ concentration was 71 mg/L and $Cl^-$ concentration was 1670 mg/L. Therefore, as the Fe(II) concentration was increased by a factor of 240 (from 1.3 to 306 mg/L), the $SO_4^{2-}$ concentration only increased by a factor of 8 (from approximately 71 to 595 mg/L). We therefore expect that the dominant effect observed through this addition experiment is the effect of increasing Fe(II), rather than the effect of increasing $SO_4^{2-}$ and/or total anions. The anion and cation concentrations in the experimental spiked solution were similar to the natural conditions occurring in the aquifer. For example, for the depths with $Fe^{2+}$ from 50–435 mg/L, the range in $SO_4^{2-}$ was 13–600 mg/L and range in $Cl^-$ was 50–9600 mg/L (Jia, 2015).

If necessary, the pH of the experimental solution was adjusted using 0.1 M NaOH or 0.1 M HCl (to 7.4 ± 0.3) to match the target pH (7.44) of the original DOM stock solution. After each Fe(II) addition, 10 mL of the Fe(II) solution was pipetted into each of two glass cuvettes. One cuvette was acidified with concentrated HCl to a pH of approximately 2, and used to determine the total dissolved Fe(II), using a HACH$^{TM}$ DR/2010 spectrophotometer via the colorimetric method (Hach ferrozine method) (Stookey, 1970). The cuvette of Fe(II) solution used for fluorescence analysis was capped tightly, and transferred out of the glove box for immediate analysis (section 3.3).

### 3.3 Fluorescence data acquisition and analysis

The fluorescence analysis and the PARAFAC modeling were described by Ishii & Boyer (2012). All fluorescence spectra were obtained by using a Horiba Aqualog® (Horiba Scientific, Edison, NJ, USA) spectrofluorometer, equipped with subtractive double excitation monochromators (Hansen et al., 2018). A 150 W ozone-free vertically mounted xenon arc lamp was used as the excitation source. Both excitation and emission were collected at a bandpass of 5 nm. Fluorescence intensities, as a function of the excitation and emission wavelengths, were measured across excitation wavelengths ranging from 240 to 800 nm in 3 nm increments; emission wavelengths, ranging from 250 to 830 nm, were measured over an integration time of 0.1 s. Water samples were analyzed in 1 cm quartz cuvettes. Between the samples, the quartz cuvette was rinsed 3 times with Milli-Q water, followed by 3 times with the sample, to reduce possible cross-contamination. If necessary, water samples were quantitatively diluted with Milli-Q water until the UV absorbance was lower than 0.2 units (at 254 nm) to minimize inner filter effects between the Milli-Q water and the water samples. The EEM spectra for each sample was obtained by subtracting the Milli-Q (blank) spectra to eliminate the Rayleigh scatter and water Ramen peak (Murphy, 2011). Fluorescence intensity within all EEM data is presented in Raman units (RU) due to the way that raw EEM spectra are corrected prior to analysis via PARAFAC modelling or calculation of associated indices. As per standard practice, raw EEMs were instrument corrected via software provided by the instrument manufacturer. Spectra were corrected for inner filter effects (Ohno, 2002), then normalized to the area under the Raman curve (Nieke et al., 1997; Stedmon et al., 2003); second order Raleigh scatter and Raman bands were excised at a bandpass of 12 nm (Bahram et al., 2006; Zepp et al., 2004), while first order Raleigh scatter was excised at a bandwidth of 50 nm to remove all spectral artefacts (Bro, 1997; Stedmon & Bro, 2008). Specifically, normalization to the area under the Raman curve (which occurs due to the inelastic scatter of light by water) contributes to instrument correction that allows for the comparison of spectra between different instruments and thus different studies. The overall fluorescence intensity (OFI) was determined for each sample by adding the fluorescence intensities across all EEMs (Poulin et al., 2014). The relative fluorescence (OFI/OFI$_0$) can be used to quantitatively determine the quenching effect of Fe(II) (Poulin et al., 2014), where OFI and OFI$_0$ represent the Fe(II) addition samples and the original groundwater sample, respectively. Similarly, the intensity of the primary peak (Peak A) at an excitation wavelength of 239 nm, and at broad emission wavelengths ranging from 380 to 460 nm, was determined for each sample, and the parameter A/A$_o$ was used to quantify the Fe(II) quenching effect on this diagnostic peak.

The established 13-component PARAFAC model of Cory & McKnight (2005) was used to fit the EEM spectra within this study. The 13 components consist of seven quinone-like fluorophores, including three oxidized quinones (Q1, Q2, and Q3), four reduced quinones (SQ1, SQ2, SQ3, and HQ), two amino acid-like components (tryptophan and tyrosine), and four remaining unknown fluorophores (Cory & McKnight, 2005). We chose to use this robust pre-resolved model, which was developed using DOM from a wide range of aquatic environments and has been subsequently applied to interpret EEMs from a large variety of aquatic systems (Jaffé et al., 2008; Larsen et al., 2010); see section 3.3 for further details. The use of this 13-component model also facilitates the derivation of the redox index (RI), calculated by summing the reduced quinone-like inputs over total quinone-like inputs from components within the model. Finally, derivation of a unique, site-specific PARAFAC model typically requires a large sample set composed of samples from a common organic matter context (Cory & McKnight, 2005; Ishii & Boyer, 2012). As the aim of this study was to capture how spectral attributes are quenched upon addition of Fe(II), rather than characterization of the underlying organic matter properties, the application of a pre-resolved model ensures that model fitting is not biased by Fe(II) addition.

To ensure that this 13-component model adequately represented the fluorescent organic matter characteristics within the sample set, the residual fluorescence remaining after the model was applied were plotted and analysed. No systematic residuals were found after fitting the EEMs to the PARAFAC model, suggesting that the model was able to represent the samples, and that Fe(II) additions did not significantly change the structure of fluorophores in the groundwater stock solution from the Kidd 2 site. The abundance of each fluorophore was quantified based on its relative contribution (%) to the total fluorescence. Additionally, commonly-used fluorescence indices, including fluorescence index (FI) (Cory & McKnight, 2005), humification index (HIX) (Ohno, 2002; Parlanti et al., 2000), the redox index (RI) (Miller et al., 2006), and freshness index (FrI, $\beta/\alpha$) (Parlanti et al., 2000; Zsolnay et al., 1999) were also quantified to provide further DOM characterization (section 4.1.3). The dataset from this study is available on Zenodo (Jia et al., 2020).

# 4 Results

## 4.1 The effect of Fe(II) quenching on EEM fluorescence

### 4.1.1 Relative fluorescence intensity (OFI/OFI$_0$)

Figure 3 shows that the fluorescence intensities of EEMs decrease with increasing Fe(II), indicating that the DOM fluorescence of the groundwater stock solution collected from the Kidd 2 site was quenched by the addition of Fe(II). Figure 4a presents the decrease in the relative fluorescence intensity (OFI/OFI$_0$) as Fe(II) increases from 1 to 306 mg/L. Approximately 60% of the fluorescence intensity found in the 1 mg/L Fe(II) experimental solution was quenched in the 306 mg/L Fe(II) experimental

solution. The fluorescence intensity decreased more rapidly at lower Fe(II) concentrations: as Fe(II) increased from 1 to 101 mg/L the OFI decreased by ~40% and as Fe(II) increased from 101 to 306 mg/L, the OFI decreased by an additional ~20%. The magnitude of quenching effect was more pronounced in this study than that performed by Poulin et al. (2014), who observed nonlinear fluorescence quenching (7% to 23%) in four different surface water samples, by addition of Fe(II) up to 1.5 mg/L, significantly lower than the Fe(II) concentrations used in this study.

Fe(III) also interacts with DOM and quenches fluorescence intensities (Ohno et al., 2007; Poulin et al., 2014). In this experiment, it was expected that some portion of Fe(II) would have oxidized to Fe(III) since the samples were directly exposed to the atmosphere when they were transferred to the sample cuvette for analysis. Nevertheless, the oxidation from Fe(II) to Fe(III) was limited due to the minimal exposure, and no visible Fe(III) colloids were observed prior to the analysis. The analysis took about 5-10 minutes for each sample; the full analysis was completed within an hour. Therefore, the quenching effect was unlikely to be caused by dynamic colloid formation. Furthermore, Poulin et al. (2014) found that almost no quenching was observed by Fe(III) from the oxidation of Fe(II) to Fe(III) at pH 6.7. Hence, the quenching effect in this experiment was believed to be primarily due to Fe(II)-DOM interactions.

### 4.1.2 Relative Peak A fluorescence intensity ($A/A_0$)

Many studies have characterized fluorescence properties of waters based on the primary peaks in EEM spectra, identified by visual inspection and/or multivariate data analysis (Chen et al., 2003; McKnight et al., 2001; Murphy et al., 2013; Shen et al., 2020; Stedmon et al., 2003; Stedmon & Bro, 2008). The positions of these peaks are believed to be linked to the organic matter properties. Coble (1996; 1990) identified five primary peaks from a visual inspection of EEMs, including humic-like Peaks A, C, and M; and protein-like Peaks B and T. We observed only one distinct humic-like fluorescence peak (Peak A) in the EEMs from W3-14. Peak A was in the UV region at an excitation wavelength of 239 nm, and at broad emission wavelengths ranging from 380 to 460 nm.

Similar to trends for relative OFI (section 4.1.1), the relative intensity of Peak A decreased by ~60% as Fe(II) increased from 1 to 306 mg/L, and over 65% of the quenching occurred below Fe(II) concentrations of 101 mg/L (Figure 4c). In addition, the position of Peak A continuously migrated toward the shorter (i.e., higher energy) emission wavelengths with a constant excitation wavelength of 239 nm and increasing Fe(II) concentration. Figure 4e presents the emission positions of Peak A along with Fe(II) concentrations at excitation 239 nm. Although a linear relationship was not observed, overall the location of fluorescence response gradually changed from 441 to 409 nm as Fe(II) increased from 1 to 306 mg/L. This result is consistent with quenching experiments conducted with Everglades F1 water samples, where Poulin et al. (2014) observed a distinct shift in the quenching locations with increasing ratio of Fe(II) to DOM.

### 4.1.3 Fluorescence intensities

Iron quenching also affects several indices that are used to quantify DOM fluorescence properties. The most common indices are the fluorescence index (FI) (Cory & McKnight, 2005), humification index (HIX) (Ohno, 2002; Parlanti et al., 2000), the redox index (RI) (Miller et al., 2006), and freshness index ($\beta/\alpha$) (Parlanti et al., 2000; Zsolnay et al., 1999). By defining the ratios of fluorescence intensity in different regions of the EEMs, indices can provide insight into the source of DOM, the degree of humification, and the relative age of the recently produced DOM.

The fluorescence index (FI) is the most widely used index that provides information about the source of organic matter. FI is defined using instrument-corrected spectra as the ratio of emission measured at 470 nm to that measured at 520 nm, both from an excitation of 370 nm (Cory & McKnight, 2005). In the absence of fluorescence quenching by other dissolved constituents, high values of FI (approximately 1.80) indicate that DOM is derived from extracellular microbial activity, whereas low values of FI (approximately 1.20) suggest that DOM comes from terrestrial plant and soil organic matter (Cory & McKnight, 2005).

Measured FI values increased from an initial 1.62 to 1.80 ($\Delta$FI = +0.18 FI units) with increased Fe(II) concentrations (Figure 4b), indicating the susceptibility of FI to the iron-quenching effect. As FI is a ratio of emission intensities, non-uniform changes in component emissions are responsible for the increase in FI values with Fe(II). Nevertheless, the effect of iron-quenching on DOM fluorescence and FI were only observed in the 1 to 101 mg/L Fe(II) concentration range. As the Fe(II) increased from 101 to 306 mg/L, the measured FI remained stable at ~1.80. Poulin et al. (2014) also observed that FI values increased more rapidly at low Fe(II) concentrations, and began levelling off approaching the maximum Fe(II) concentrations that they studied, 1.5 mg/L. A stable value of FI was not reached in Poulin et al. (2014) Fe(II) addition experiment, probably because 1.5 mg/L of added Fe(II) did not saturate all available DOM ligands.

The humidification index (HIX) is defined as the peak area under the emission spectra from 435-480 nm, divided by the peak area from 300-345 nm + 435-480 nm, at an excitation of 254 nm and typically ranges from 0-1 (Ohno, 2002). Higher values of HIX (closer to 1) indicate greater humic content and extent of humidification. HIX values decreased with the addition of Fe(II) (from about 0.93 to 0.84) indicating that the emission spectra of fluorescence shifted toward shorter wavelengths (Figure 4d). Moreover, the decrease in HIX occurred in two phases with increasing Fe(II), as represented by a change in slope. The steeper decrease of approximately 4.5% in HIX was observed as the Fe(II) concentration changed from 1 to 72 mg/L. Above 72 mg/L Fe(II), HIX only decreased 2.9%. The results indicated that changes in emission spectra were therefore more sensitive to relatively low Fe(II) concentrations.

The freshness index (FrI or $\beta/\alpha$) is defined as the ratio of emission at 380 nm ($\beta$) divided by maximum emission between 420 and 435 nm ($\alpha$), all at an excitation of 310 nm (Parlanti et al., 2000; Wilson & Xenopoulos, 2009). It is a measure of the

proportion of recently produced DOM, where β represents freshly produced DOM and α represents more decomposed DOM (Parlanti et al., 2000; Wilson & Xenopoulos, 2009). A higher FrI indicates more recently created DOM, with values >1 indicating freshly released DOM and lower values (0.6-0.7) correspond to older DOM with a predominantly terrestrial source (Parlanti et al., 2000). Overall, the freshness index ranged between 0.72 to 0.84 and increased with Fe(II), except for the slight decrease seen at the Fe(II) concentration of 101 mg/L (Figure 4f). Similar to trends for HIX, the more rapid change of 10.4% occurred as the Fe(II) concentration ranged from 1 to 72 mg/L and a more gradual change of 6.3% occurred as the Fe(II) concentration ranged between 72 and 306 mg/L.

## 4.2 The effect of Fe(II) quenching on PARAFAC modeling and component distribution

Of the 13 components identified by Cory & McKnight (2005), seven components were identified as quinone-like organic components (including three oxidized quinones, Q1, Q2, and Q3 and four reduced quinones, SQ1, SQ2, SQ3, and HQ), based on the similarity of the positions and relative intensities of the component excitation peaks compared to the absorbance and excitation peaks of model quinones. Two components were defined as resembling amino acid fluorophores (C8, tryptophan, and C13, tyrosine). The remaining four components (C1, C3, C6, and C10) have not been associated with any class of molecule. The three most abundant components were Q1, Q2, and Q3; together they contributed 49-52% to the total fluorescence (Figure 5). The excitation and emission spectra of the 13 components are included in our dataset published on Zenodo.

The fluorescence intensity of each component peaked from 1 to 44 mg/L Fe(II) and then steadily decreased as Fe(II) increased to 306 mg/L (Figure 5a). For all components except tryptophan (C8), the fluorescence at 306 mg/L Fe(II) was less than the fluorescence at 1 mg/L Fe. As Fe(II) increased from 1 to 306 mg/L, the total fluorescence intensity of the 13 components decreased by approximately 50%, from 207 to 101 RU. However, the relative proportions of the components were relatively stable. The maximum deviation was seen in C6 (unknown classification) which decreased in relative proportion from 9 to 5% as Fe(II) increased from 1 to 306 mg/L.  For the other components, the changes in proportions were restricted to be within ±3%. We conclude that the proportions of the 13 components are relatively insensitive to the Fe(II) concentration for the DOM in the experimental stock solution from the Kidd 2 site.

The trends in component distribution can be further evaluated using the redox index (RI), which is calculated as the sum of reduced quinone-like inputs over the total quinone-like input. RI measures the oxidation state of the DOM and redox reactivities (Miller et al., 2006; Mladenov et al., 2008). The RI can be used to determine whether the quinone-like components within the DOM are more reduced (RI closer to 1) or more oxidized (RI closer to 0). A shift in the RI usually indicates changes in the redox status. The total oxidized (Q1, Q2, and Q3) and reduced (SQ1, SQ2, SQ3, and HQ) quinones changed their contributions to total fluorescence from 52 to 50% and from 21 to 20%, respectively. The small changes for both oxidized and reduced quinones are responsible for the relatively stable values of RI, which slightly decreased from 0.30 to ~0.285 (Figure 5b).

## 5 Discussion

Ferrous iron is responsible for the observed quenching of DOM fluorescence. Both the total fluorescence intensity of EEMs and Peak A intensity decreased with increasing Fe(II). The likely mechanism accounting for the quenching is formation of organic-metal complexations, which occupy the fluorescent sites and efficiently decrease the fluorescence intensity of fluorophores (Senesi, 1990; Waite & Morel, 1984). The non-linear quenching of the fluorescence intensity with Fe(II) suggests, following Senesi (1990), a static quenching mechanism. However, our experiment does not allow us to identify quenching mechanisms. While the maximum ratio of Fe(II) to DOM (mg/L per mg/L) was approximately 0.4 in Poulin et al (2014), it is much larger in our study, with a molar Fe to C (as DOC) ratio of approximately 7. While earlier work by Senesi (1990) suggests that quenching primarily depends upon the fraction of DOM ligands that are complexed to Fe(II), our study, with a great excess of Fe(II) over C in DOM, could involve other mechanisms. Poulin et al. (2014) found that Fe(II)-quenching of fluorescence ranged from 7% to 23% in four different hydrophobic acid (HPoA) fractions and surface water samples, with DOC concentrations ranging from 2.3 to 5.0 mg/L, and the degree of quenching was not related to DOM concentrations. In our study, both total fluorescence intensity and Peak A intensity decreased by approximately 60% as Fe(II) increased from 1 to 306 mg/L. The differences in the impact of Fe(II) on fluorescence quenching between the representative groundwater from Kidd 2 site and those reported by Poulin et al. (2014) could be related to the much higher Fe(II) background in the organic-rich aquifer and groundwater at the Kidd 2 site. It should be noted that the DOC concentration in the Kidd 2 groundwater was 10.7 mg/L, significantly greater than that in the previous study (Poulin et al., 2014).

Poulin et al. (2014) mainly examined the effect of Fe(II) addition to terrestrial-derived fresh surface water with undetectable Fe(II) levels. In contrast, the DOM in the stock solution collected from the Kidd 2 site is hypothesized to be derived from microbial sources and may respond to high Fe(II) concentrations differently than freshwater terrestrial-derived DOM. The quenching mechanism for this DOM is not well understood.

Similar metal quenching of humic-like peaks has been observed by other researchers. Ohno et al. (2007) conducted experiments on the impact of Fe(III) and Al(III) addition to the deciduous water-soluble organic matter (WSOM) fluorescence spectra. This result showed that the fluorescence intensity was quenched by about 30% in the presence of 25 μM (1.4 mg/L) Fe for Peak A (Ohno et al., 2007).

The fluorescence indices FI, HIX, β/α, provided evidence of the susceptibility to Fe(II) quenching, while RI was relatively insensitive to Fe(II) concentration. FI values increased by approximately 0.18 units from the addition of 300 mg/L of Fe(II), and shifted towards greater microbial-derived origin. This relatively small change may not affect the utility of FI towards inferring DOM origin. Moreover, FI values are more sensitive at low Fe(II) concentrations and they gradually reach a plateau once Fe(II) is above 101 mg/L, as all available DOM ligand sites are likely fully occupied by Fe(II)-DOM interactions. This

result also supports a static quenching mechanism, since quenching does not depend on the Fe(II) concentration (Senesi, 1990). Similar to FI, both HIX and β/α show more pronounced changes at low Fe(II) concentrations (Fe(II) < 72 mg/L). The decrease in HIX and increase of the β/α are consistent, as they both indicate that the DOM shifted to more freshly produced, with a higher H:C ratio and less polycondensation (Ohno, 2002). Nevertheless, the HIX and the β/α are not likely to reach a constant value with Fe(II) concentration. Therefore, conclusions about organic matter origin based on humidification and freshness indices must consider the DOM sensitivity to Fe(II) concentration when reporting values.

In the 13 component PARAFAC model (Cory & McKnight, 2005), all components except C8 were quenched in terms of their fluorescence intensities. Nevertheless, changes in their "relative" component distribution were relatively small (within ±4% contribution to total fluorescence). This result was consistent with Poulin et al.'s (2014) observation, where changes in the component distributions were within ±3% of the total fluorescence in their 13-component PARAFAC model. The small or negligible change in component distribution results in the relatively constant RI. Although few fluctuations were observed in the Fe(II) addition experiment, the overall trend of the RI is stable; thus, it can be used as a reliable index to infer the redox condition of the aquatic environment. Still, the component distribution may vary in different PARAFAC models. There were more pronounced distribution changes in a 7-component model than in a 13-component model (Poulin et al., 2014). This suggests that the degree of quenching should always be associated with fluorophore classification. Besides fluorophore classification, DOM composition is another important factor that controls the quenching characterization. Ohno (2009) found that quenching of fluorescence intensity varies depending on the DOM composition. In their 3-component PARAFAC model, Fe(III) quenching was observed in all three components for the deciduous WSOM sample. For the coniferous WSOM sample however, only component 1 and 2 were quenched, and component 3 increased slightly with the initial addition of Fe(III), and decreased with further Fe(III) additions.

While our study shows that high Fe(II) concentrations can influence fluorescence properties of a representative deltaic groundwater, it is difficult to generalize our results to other terrestrial waters with different DOM compositions without further analyses. We observed non-uniform quenching as a function of Fe concentration in this study, with smaller increases in quenching at concentrations above 100 mg/L. We note that both the DOM and Fe(II) concentrations that we examined were higher than those examined by Poulin et al (2014).

**6 Summary**

This study demonstrates the quenching effect of Fe(II) in organic-rich anoxic groundwater from the Kidd 2 aquifer in the Fraser River Delta, Richmond, BC, where Fe(II) concentrations range from 1 to 300 mg/L and the DOM concentration is ~10 mg/L. In our experiments, total fluorescence intensity decreased by approximately 60% as Fe(II) increased from 1 to 300 mg/L. While our results are likely applicable to similar deltaic groundwaters, further analyses are required to quantify the quenching effect on fluorescence indices in other terrestrial waters (for example, surface waters with sufficiently high Fe

concentrations such that quenching is likely). In this study, FI values tended to shift to be more autochthonous in origin with increasing Fe(II), but the small changes are unlikely to invalidate conclusions about the source of DOM. Changes in the humidification index and freshness index both indicated that the addition of Fe(II) would shift these two indices towards more recently produced DOM. Therefore, the sensitivity of these indices should be evaluated when water samples contain Fe(II). The non-linear relationship between the indices and the Fe(II) can be seen in all of the indices, especially the FI. Although the intensities of all 13 components varied as a function of Fe(II), the relatively stable component distribution suggests that the Fe(II) quenching effect has a negligible effect on the 13-component PARAFAC model. As a result, the PARAFAC model can be a reliable method for obtaining information about DOM composition in their relative distributions and redox status via the RI.

## Data availability

The data presented in this manuscript has been published on Zenodo at http://doi.org/10.5281/zenodo.3737108 (Jia et al., 2020).

## Author contribution

KJ, AJ, and RB designed experiments and KJ conducted experiments. KJ and AJ performed data analysis. KJ and CM prepared figures. CM compiled and archived the data. KJ and CM prepared the manuscript with contributions from AJ and RB.

## Acknowledegments

We thank Mark Bolton for assistance with sample analysis, and two anonymous reviewers for their helpful feedback that improved the manuscript.

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

**Figures**

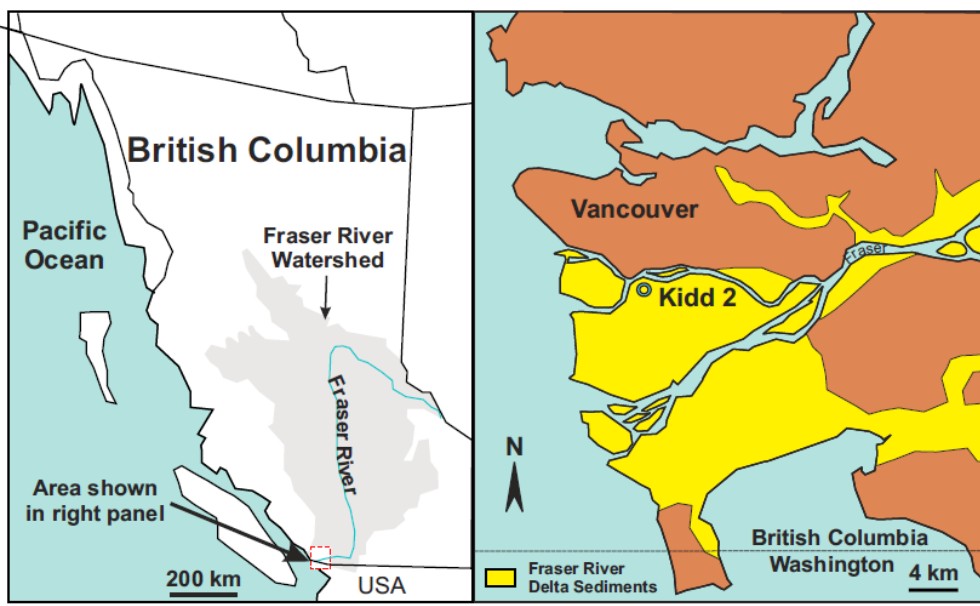

**Figure 1:** The Kidd 2 site is located in the Fraser River delta in southwest British Columbia.

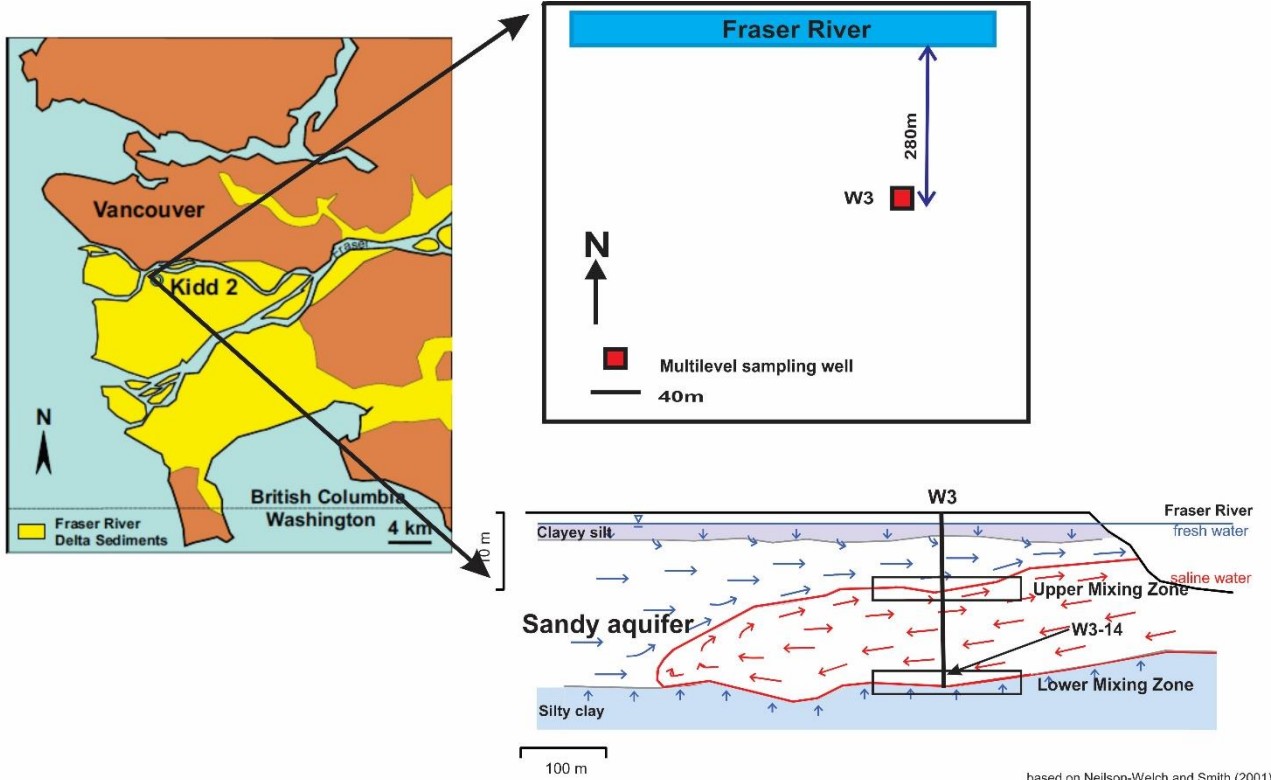

**Figure 2:** Plan view of the W3 well location and cross-section of the saline wedge, after Neilson-Welch and Smith (2001).

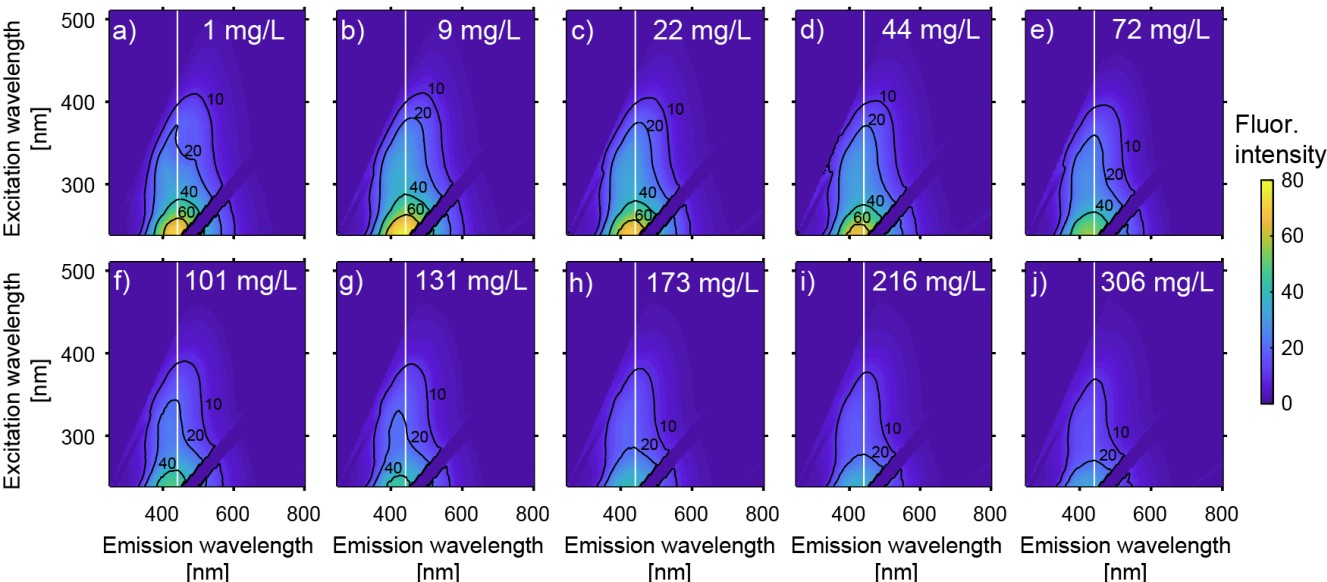

**Figure 3:** Excitation-emission matrices (EEMs) of groundwater from Fraser River aquifer over a range of Fe(II) concentrations. The Fe(II) concentration for each sample is indicated in mg/L in the top right of each plot, from a) 1 mg/L to j) to 306 mg/L. To distinguish changes in the center of the primary peak (peak A), a vertical dashed line at an emission wavelength of 441 nm is shown (the peak emission wavelength at excitation 239 nm for the 1 mg/L solution, see Figure 4). The fluorescence intensities are reported in Raman units (RU).

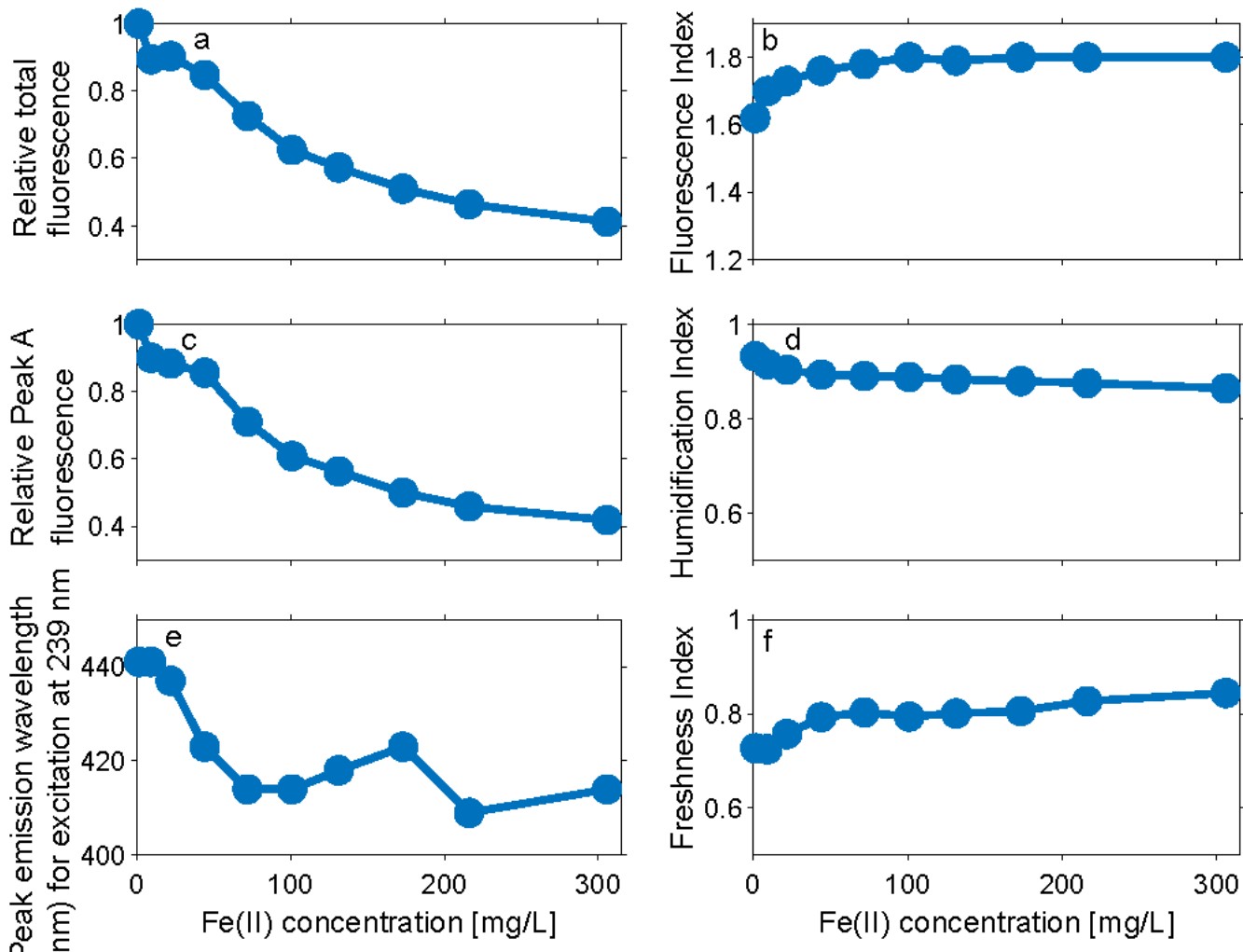

**Figure 4:** Effect of varying Fe(II) concentrations at pH 7.4 on: a) relative total fluorescence intensity (OFI/$OFI_0$); c) relative fluorescence intensity of Peak A (A/$A_0$) and e) peak fluorescence emission wavelength (nm) at excitation at 239 nm (peak A). Effect of varying Fe(II) concentration at pH 7.4 on various indices: b) fluorescence index (FI), d) humidification index (HIX), and f) freshness index (FrI). The results show that the fluorescence intensities, indices and peak emission wavelength change as Fe(II) is increased from 1 to 306 mg/L.

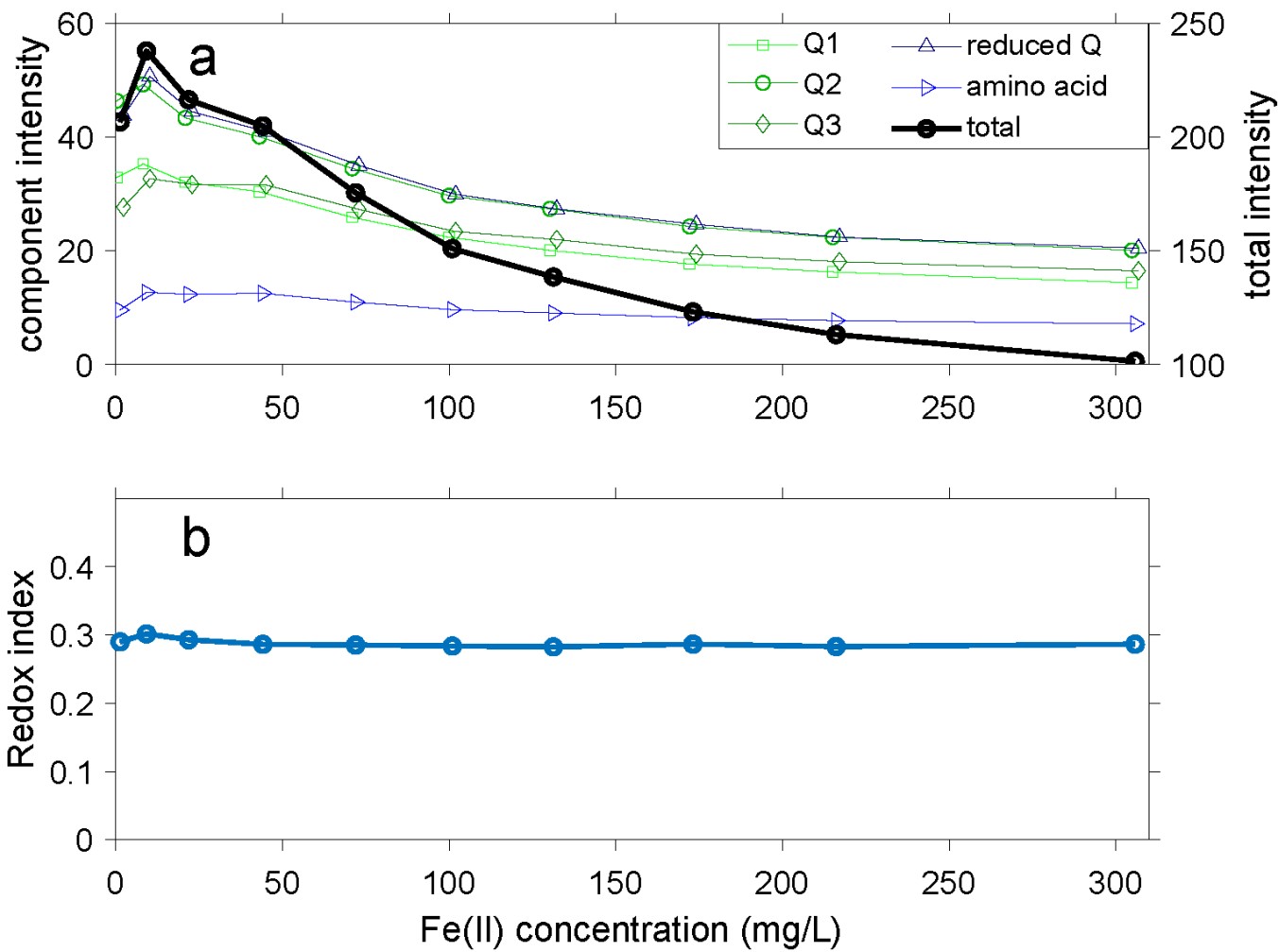

**Figure 5:** a) Intensity of components at varying Fe including quinones Q1, Q2, and Q3 (green), reduced quinones (reduced Q, calculated as the sum of SQ1, SQ2, SQ3, and HQ), amino acid (tryptophan and tyrosine) and total intensity, and b) redox index
(RI) in the presence of varying Fe(II). In a) the Fe concentrations have been slightly offset to better show overlapping symbols.