# Peer review of "Technical note: Effects of iron(II) on fluorescence properties of dissolved organic matter at circumneutral pH"

_Hydrology and Earth System Sciences, 2020_

## Referee Comment (RC1) · Anonymous Referee #1 · 9 Jun 2020

The authors present a characterisation study on the effect of dissolved Fe(II) on the fluorescence properties of organic matter dissolved in groundwater. The study is novel and the experimental observations are well performed. This makes the manuscript suitable for publication. However, the paper does not fit well to the HESS audience as this audience is less focused on analytical geochemistry. The paper refers to the second scope of HESS: it is noted that one groundwater sample is used for characterisation which makes spatial or temporal characterisation not that strong. To me, the paper would better fit in Appl Geochem or Chem Geol or so. I leave it to the editor to decide on this.

Having said this, I have three major comments. Other comments are more specific comments that refer to these three major comments are annotated in the pdf version of the manuscript.

First, the relevance of high Fe in groundwater is not illustrated. The authors use Fe concentrations up to 300 mg/L. This is a very high value that is rarely found in groundwater, especially when it has a neutral pH. The authors should thus present their paper within the framework of high-fe groundwater: what has been observed and for which hydrological conditions.

Second, the authors discuss that the effect of Fe(II) on the fluorescence properties is due to aqueous complexing but do not elaborate on this. They should pay more attention to this and discuss why this effect is so strong for Fe(II). Might one expect a similar effect for dissolved Ca which is more omnipresent. If not, what makes Fe(II) so unique besides Fe(III) and Al(III)?

Third, as said before, the paper is oriented towards analytical chemistry. To make it understandable for the HESS audience, they should explain specific details of the technique, etc. Now, the paper will not reach the HESS audience as the relevance is not well illustrated (comment 1) and the text is too technical.

Please also note the supplement to this comment:
https://www.hydrol-earth-syst-sci-discuss.net/hess-2020-150/hess-2020-150-RC1-supplement.pdf

**Supplement:**

[revised manuscript text omitted]

---

## Author Comment (AC1) · 10 Jun 2020

Dear Editor and Reviewer 1,

We thank Reviewer 1 for their thorough and constructive review of our manuscript and the positive feedback that the "study is novel and the experimental observations are well performed" and that the manuscript is "suitable for publication." We plan to provide a full response to this review once we have received all of the reviews and the Editor's decision.

In the meantime, we wish to address the reviewer's comments that "the paper does not fit well to the HESS audience as this audience is less focused on analytical geochemistry." The scope of HESS includes "the role of physical, chemical, and biological processes in the cycling of continental water in all its phases, including dissolved and particulate matter, at all scales." Along these lines, "biogeochemical processes" is a subject area for the journal. The manuscript type we submitted (technical note) is for papers relating to "experimental and theoretical methods and techniques which are relevant for scientific investigations within the journal scope."

This technical note provides new methodological insights for the interpretation of fluorescence spectroscopy data (excitation-emission matrices, EEMs) in groundwater samples containing significant $Fe^{2+}$. Fluorescence spectroscopy techniques and EEMs are frequently applied to characterize dissolved organic matter and biogeochemical processes in a wide range of continental waters.

To support our argument that the manuscript falls within the scope and readership of the journal, we have provided a references list on the following page containing six articles published in HESS that include dissolved organic matter characterization using EEMs. Additionally, because HESS is an open access journal, the article will be able to reach the widest possible audience compared to traditional subscription-based geochemistry journals.

We are currently preparing a separate manuscript focused on full geochemical characterization of groundwater and sediments across the saline-freshwater gradient at the Kidd 2 site, where the groundwater sample used in this technical note was collected. In this follow-up paper we will show that the highly elevated $Fe^{2+}$ is observed due to reductive dissolution of iron oxide minerals via organic matter oxidation. The results of this technical note are required to interpret the fluorescence spectroscopy measurements in the context of the wide range of $Fe^{2+}$ concentrations observed at the site (10 to 300 mg/L). We decided to publish this technical note separately to allow the site characterization paper to be more focussed.

Thank you again for your helpful review. We look forward to providing a full response to your comments.

Sincerely,
Cara Manning on behalf of coauthors Kun Jia, Ashlee Jollymore and Roger Beckie

**References:**

Bernal, S., Lupon, A., Catalán, N., Castelar, S., & Martí, E. (2018). Decoupling of dissolved organic matter patterns between stream and riparian groundwater in a headwater forested catchment. *Hydrology and Earth System Sciences*, *22*(3), 1897–1910. https://doi.org/10.5194/hess-22-1897-2018

Bernard-Jannin, L., Binet, S., Gogo, S., Leroy, F., Défarge, C., Jozja, N., et al. (2018). Hydrological control of dissolved organic carbon dynamics in a rehabilitated *Sphagnum -* dominated peatland: a water-table based modelling approach. *Hydrology and Earth System Sciences*, *22*(9), 4907–4920. https://doi.org/10.5194/hess-22-4907-2018

Broder, T., Knorr, K.-H., & Biester, H. (2017). Changes in dissolved organic matter quality in a peatland and forest headwater stream as a function of seasonality and hydrologic conditions. *Hydrology and Earth System Sciences*, *21*(4), 2035–2051. https://doi.org/10.5194/hess-21-2035-2017

Graeber, D., Goyenola, G., Meerhoff, M., Zwirnmann, E., Ovesen, N. B., Glendell, M., et al. (2015). Interacting effects of climate and agriculture on fluvial DOM in temperate and subtropical catchments. *Hydrology and Earth System Sciences*, *19*(5), 2377–2394. https://doi.org/10.5194/hess-19-2377-2015

Pennino, M. J., Kaushal, S. S., Mayer, P. M., Utz, R. M., & Cooper, C. A. (2016). Stream restoration and sewers impact sources and fluxes of water, carbon,and nutrients in urban watersheds. *Hydrology and Earth System Sciences*, *20*(8), 3419–3439. https://doi.org/10.5194/hess-20-3419-2016

Shatilla, N. J., & Carey, S. K. (2019). Assessing inter-annual and seasonal patterns of DOC and DOM quality across a complex alpine watershed underlain by discontinuous permafrost in Yukon, Canada. *Hydrology and Earth System Sciences*, *23*(9), 3571–3591. https://doi.org/10.5194/hess-23-3571-2019

---

## Referee Comment (RC2) · Anonymous Referee #2 · 24 Jun 2020

This study presents results on the influence of Fe(II) on the fluorescence properties of DOM. The concept is novel and addresses a relevant scientific question. However, the results fail to support the conclusions, specifically the PARAFAC modeling results. My recommendation is that the paper should be reconsidered after major revisions. I have three major comments and the rest are addressed in the comments of the attachment.

1. Papers with PARAFAC modeling results typically publish the modeled components and sometimes also the modeled excitation and emission spectra, or at least include them for closer inspection in supplementary material. I'm not sure familiar HESS audience is with PARAFAC, but seeing the modeled components help with understanding.

[Figure]

There was also no mention made of your validation techniques. Validating a model can be very subjective. What steps did you take to resolve this dataset into 13 components? I found it difficult to comment on your PARAFAC results without first knowing how you arrived at your modeled results.

2. I strongly recommend defining optical measurements (absorbance and fluorescence), then describe why they are useful to study DOM, and then define the specific optical properties (FI, HIX, etc.) that were used diagnostically to describe DOM and Fe interactions in this study.

3. This paper could be vastly improved by a more complete and thorough literature review. Some statements were not attributed appropriately/completely. See comments within doc.

Please also note the supplement to this comment:
https://www.hydrol-earth-syst-sci-discuss.net/hess-2020-150/hess-2020-150-RC2-supplement.pdf

**Supplement:**

[revised manuscript text omitted]

---

## Author Comment (AC2) · 6 Oct 2020

**Response to Referee 1 comment on "Technical note: Effects of iron(II) on fluorescence properties of dissolved organic matter at circumneutral pH"**

Please note the reviewer's comments are in *purple italic Verdana font* and our responses are in Times New Roman, with proposed revisions to the text in blue.

**Anonymous Referee #1**
**General comments:**

*The authors present a characterisation study on the effect of dissolved Fe(II) on the fluorescence properties of organic matter dissolved in groundwater. The study is novel and the experimental observations are well performed. This makes the manuscript suitable for publication. However, the paper does not fit well to the HESS audience as this audience is less focused on analytical geochemistry. The paper refers to the second scope of HESS: it is noted that one groundwater sample is used for characterisation which makes spatial or temporal characterisation not that strong. To me, the paper would better fit in Appl Geochem or Chem Geol or so. I leave it to the editor to decide on this.*

We addressed this comment in our initial response to Referee 1 (Author Comment 1, available at https://doi.org/10.5194/hess-2020-150-AC1). Quoting from Author Comment 1:

"The scope of HESS includes 'the role of physical, chemical, and biological processes in the cycling of continental water in all its phases, including dissolved and particulate matter, at all scales.' Along these lines, 'biogeochemical processes' is a subject area for the journal. The manuscript type we submitted (technical note) is for papers relating to 'experimental and theoretical methods and techniques which are relevant for scientific investigations within the journal scope.'… Additionally, because HESS is an open access journal, the article will be able to reach the widest possible audience compared to traditional subscription-based geochemistry journals."

The referee's comment *"one groundwater sample is used for characterisation which makes spatial or temporal characterisation not that strong"* points to a misapprehension. The purpose of the manuscript is not to characterize DOM at the site from which the sample (stock solution) was collected. Rather, it is to understand how ferrous iron affects the fluorescence properties of DOM in groundwater. We selected our single stock solution because it was typical of the deltaic aquifer groundwaters that can contain high ferrous iron concentrations. We will revise the manuscript to make this objective clearer.

**Original:**
In this study, up to 300 mg/L Fe(II) was added to DOM-containing groundwater to assess the influence of Fe(II) on the fluorescence properties of DOM.

**Proposed revision:**
The objective of this study was to assess the influence of high concentrations of Fe(II) on the fluorescence properties of DOM by titrating up to 300 mg/L (5.4 mM) Fe(II) into groundwater collected from a deltaic aquifer in Richmond, British Columbia, Canada.

*First, the relevance of high Fe in groundwater is not illustrated. The authors use Fe concentrations up to 300 mg/L. This is a very high value that is rarely found in groundwater, especially when it has a neutral pH. The authors should thus present their paper within the framework of high-fe groundwater: what has been observed and for which hydrological conditions.*

Our study is motivated by the high, naturally-occurring iron concentrations we have observed in groundwaters in deltaic sediments of south Asia, and at the Kidd 2 field site in Vancouver Canada. High iron concentrations are common in deltaic sediments, although the over 300 mg/L found in parts of the Kidd 2 site is unusual for circumneutral groundwater. Similarly, organic contamination (fuel spills, landfills) can also drive iron reduction and produce high iron concentrations under circumneutral pH conditions. We have revised our motivational statement to make this clearer to the reader.

**Original:**
The fluorescence quenching effect in Fe(II)-rich groundwater is still poorly understood and warrants further investigation, given the prevalence of high DOM and Fe(II) in groundwater in deltaic sediments (Bolton & Beckie, 2011) and contaminated sites (Christensen et al., 2001). Fe(II) is often present in groundwater in organic-rich settings where the oxidation of organic matter is coupled to solid-phase Fe(III) reduction, dissolving Fe(II) into groundwater. For example, 1.5-10 mg/L Fe(II) in groundwater is commonly observed in the organic-rich groundwaters of the Bengal Basin (Harvey et al., 2002).

**Proposed revision:**
The fluorescence quenching effect in Fe(II)-rich groundwater is still poorly understood and warrants further investigation, given the prevalence of high DOM and Fe(II) in groundwater in deltaic sediments (Bolton & Beckie, 2011) and sites contaminated with organics, for example, from landfills or fuel spills (van Breukelen & Griffioen, 2004; Christensen et al., 2001; Heron et al., 1994). In most instances, these high Fe(II) groundwaters are found when the oxidation of organic matter is coupled to solid-phase Fe(III) reduction, dissolving Fe(II) into groundwater at circumneutral pH. For example, 1.5-10 mg/L Fe(II) in groundwater is commonly observed in the organic-rich groundwaters of the Bengal Basin (Harvey et al., 2002), and up to 90 mg/L Fe(II) has been observed in landfill leachate in the Netherlands (van Breukelen & Griffioen, 2004).

*Second, the authors discuss that the effect of Fe(II) on the fluorescence properties is due to aqueous complexing but do not elaborate on this. They should pay more attention to this and discuss why this effect is so strong for Fe(II). Might one expect a similar effect for dissolved Ca which is more omnipresent. If not, what makes Fe(II) so unique besides Fe(III) and Al(III)?*

Our intention is not to determine the mechanisms of *why* Fe(II) may affect the fluorescence properties of groundwater. While this is a legitimate topic of inquiry, it is beyond the scope of our study, which is empirically focused on characterizing the relationship for Fe(II) specifically. We hope this study motivates other research groups to characterize the effects of other cations such as $Ca^{2+}$ on fluorescence properties and to compare their results with the results in this paper. No revisions made.

*Third, as said before, the paper is oriented towards analytical chemistry. To make it understandable for the HESS audience, they should explain specific details of the technique, etc. Now, the paper will not reach the HESS audience as the relevance is not well illustrated (comment 1) and the text is too technical.*

We will add some additional text to the Introduction explaining the methodology.

**Original manuscript (line 28):**
Fluorescence spectroscopy has been widely used to characterize the properties of organic matter as it is highly sensitive to the structures and functional chemistry of aquatic organic matter (Fellman et al., 2010).

**Proposed revision:**
Fluorescence spectroscopy has been widely used to characterize the properties of dissolved organic matter as it is highly sensitive to the structures and functional chemistry of aquiatic organic matter (Fellman et al., 2010; Helms et al., 2008; Stedmon et al., 2003; Weishaar et al., 2003). In this method, light at a known wavelength (the excitation wavelength) is passed through a sample, and the absorbance at that frequency and fluorescence (emission) at other frequencies is measured. Such spectra can be used to derive commonly utilized fluorescence indices that correlate to specific forms of organic matter (Aiken, 2014; Coble et al., 2014; Hudson et al., 2007; Murphy et al., 2013). These indices include the fluorescence index (FI), which is calculated as the ratio between the emission at 470 nm to that at 530 nm at an excitation wavelength of 370 nm and relates to the concentration of aromatic, microbially derived lignin-like organic matter (McKnight et al., 2001). Additionally, an excitation-emission matrix (EEM) is prepared by systematically repeating the measurements at a range of different excitation and emission wavelengths. These measurements are highly sensitive to the structures and functional chemistry of aquatic organic matter, which determine the unique pattern of peaks present within the EEM spectra (Aiken, 2014; Coble, 1996; Coble et al., 2014; Fellman et al., 2010). Due to the complexity of the EEM spectra obtained from each unique sample, a number of statistical methods have been used to decompose EEM spectra and relate emission patterns to functional chemistry of organic matter within a sample. Parallel factor analysis (PARAFAC) is a commonly utilized statistical means of compartmentalizing EEM spectra into discrete peaks that may then be compared to broad organic matter classes (Bro, 1997; Murphy et al., 2013; Stedmon & Bro, 2008).

**Comments from PDF**
**Introduction:**
*Line 29 Omit space.*
*Line 29 This is not a nevertheless.*
These errors have been removed, see revised text in response to the previous comment.

*Line 35 does not belong here; move to end of INTRO.*
Agreed, moved.

*Line 57 better formulate an objective or research question*

Agreed. In the last paragraph of the Introduction, we now emphasize that the research question that we are assessing is the effect of high Fe(II) concentrations on groundwater that is representative of deltaic groundwaters which have extraordinarily high Fe(II) concentrations.

**Original**
In this study, up to 300 mg/L Fe(II) was added to DOM-containing groundwater to assess the influence of Fe(II) on the fluorescence properties of DOM.

**Proposed revision:**
The objective of this study was to assess the influence of high concentrations of Fe(II) on the fluorescence properties of DOM by titrating up to 300 mg/L (5.4 mM) Fe(II) into groundwater collected from a deltaic aquifer in Richmond, British Columbia, Canada.

**2 Study area**
*Line 68 I cannot see how a wedge enters an aquifer. The wedge is the result.*
Line 70 *meaning regional gradient unclear. Groundwater gradient of hydraulic heads or surface water gradient of water level?*
Line 71 *it is not according to the scope of the article, but this description is not that clear in terms of density-driven flow versus hydraulic pressure driven flow. As HESS is a hydrological journal, a better description of the hydrological forces should be given. Is there influence of tide?*
*Seems to me that reference to Jia 2015 should be made as well.*
Line 76 *put reference to figure at the start of this paragraph*

In response to these four comments, we have corrected the wording to make it clear that inflowing water forms a wedge within the aquifer. We changed "regional gradient" to "regional hydraulic gradient." The cited reference Neilson-Welch and Smith (2001) describes the density-dependent flow well whereas Jia (2015) focuses on the biogeochemistry. We do not wish to elaborate on the hydraulics of the site as density-dependent flow is complex and our concern is only that the groundwater collected at the site is representative of deltaic groundwaters with high DOM.

**Original**
At the site, a wedge of denser, saline ocean water enters the aquifer in the hyporheic zone at the river bottom, flows inland along the base of the aquifer to a maximum distance of approximately 500 m inland where it overturns flows back towards the river under a regional gradient from freshwater recharged inland (Neilson-Welch & Smith, 2001).

**Proposed revision:**
At the site, denser, saline ocean water enters the aquifer in the hyporheic zone at the river bottom, flows inland along the base of the aquifer to a maximum distance of approximately 500 m inland where it overturns flows back towards the river under a regional hydraulic gradient from freshwater recharged inland (Neilson-Welch & Smith, 2001), forming a wedge of saline water in the aquifer (Figure 2).

Line 76 *I am puzzled by this high Fe concentration. Provide more details on pH and complexing anions as SO4 and HCO3, etc. I also would like to see saturation indices for minerals as FeCO3 mentioned. And does SO4 reduction play a role or so.*

The biogeochemistry of the site is well described in Jia (2015), who uses a reactive transport model to examine the biogeochemical processes and reconcile them with the data. A detailed description of the site biogeochemistry is beyond the scope of this manuscript. We will cite Jia (2015) on this line to make the connection clearer.

**Methodology**
Line 80 *where does the stock solution come from? is it the groundwater sample?*
Yes, the stock solution is the untreated groundwater. We will add the following text as the first sentence of section 3.1.

**Proposed revision:** For the measurements in this study, we used a single stock solution of natural DOM-containing groundwater, to which we added (titrated) increasing concentrations of Fe(II).

Line 86 *teflon is rather air-permeable and I do not know about duct tape but it is a plastic and thin. Can the authors guarantee no aeration at all during 30 days storage time? Seems to me long.*

We have clarified that it was a "Teflon-lined plastic cap." The relatively inert teflon was directly in contact with the sample but there was an additional plastic barrier (the cap) between the sample and the atmosphere. We cannot guarantee that no aeration occurred, but it is likely minimal. The waters have high partial pressures of $CO_2$ and methane. What headspace may be present in the bottle is likely to be dominated by these exsolved gases. Some post-sampling alteration is tolerable, since our goal is not to characterize the groundwater at the Kidd 2 site, but to examine the effect of high Fe on the fluorescence properties of natural DOM that is representative of deltaic groundwaters.

We have revised this paragraph to clarify that the groundwater was selected as a stock solution and is not a "sample." The word "sample" may incorrectly imply that we are interested in characterizing the Kidd 2 site groundwater. Also, the Fe(II) addition experiment and fluorescence analysis were conducted within ~14 days after sample collection. The 30 days period is a conservative estimation for all laboratory analysis and data processing. The manuscript has been revised to clarify the sample handling and analysis procedures.

**Original**
Groundwater was filtered through 0.45 μm cellulose filters, then stored in a 1 L amber glass bottle- with a Teflon-lined cap, without acidification. The bottle was filled with no headspace and duct tape was used to further seal the sample and minimize the oxidation of Fe(II). The collected sample was refrigerated at 4°C until analysis (within 30 days).

**Proposed revision**
The groundwater was filtered through 0.45 μm cellulose filters, then stored in a 1 L amber glass bottle with a Teflon-lined plastic cap, without acidification. The bottle was filled with no headspace and duct tape was used to further seal the sample and minimize the oxidation of Fe(II). The collected 1 L stock solution was refrigerated at 4°C until fluorescence analysis (within ~14 days). Although some degradation of the DOM may have occurred during the holding period, we expect that this would not significantly affect our conclusions as our intention was to determine how Fe(II) addition affects the fluorescence properties of DOM, rather than to characterize the properties of DOM at the Kidd 2 site.

Line 95 *I assume gaseous? say so.*
Changed to add the word gaseous.
**Original** … which maintains $O_2$ levels of …
**Proposed revision:** …which maintains gaseous $O_2$ levels of…

Line 112, Line 251 *first word so in full: Iron(II)*
Agreed. Changed to **Ferrous iron**

Line 115 *are = were*
Agreed. Changed to were.

Line 131 *turn "was" to regular script*
Changed.

Line 141: *please provide more info on goodness-of-fit criteria. How was "no obvious residuals" assessed?*
This comment points to a misapprehension which was held by both Referees. Our objective was not to develop a model to characterize the fluorescence of the site groundwater, but, rather characterize the effects of Fe(II) on the fluorescence properties of the experimental stock solution (representative of deltaic groundwater). Our original submission was not sufficiently clear on this point. We have clarified this point in the introduction and throughout the manuscript.

**Proposed revision to Introduction:**
We identified the degree of quenching by Fe(II) based on the excitation-emission matrix (EEMs) regions and peaks. In this study we fit EEM spectra to a previously derived 13-component PARAFAC model (Cory & McKnight, 2005). We chose to use this robust PARAFAC model, which was developed using DOM from a wide range of aquatic environments and has been subsequently applied to interpret EEMs from a large variety of aquatic systems (Jaffé et al., 2008; Larsen et al., 2010). The use of this 13-component model also facilitates the derivation of the redox index (RI), calculated by summing the reduced quinone-like inputs over total quinone-like inputs from components within the model. Finally, derivation of a unique, site-specific PARAFAC model typically requires a large sample set composed of samples from a common organic matter context (Cory & McKnight, 2005; Ishii & Boyer, 2012). As the aim of this study

was to capture how spectral attributes are quenched upon addition of Fe(II), rather than characterization of the underlying organic matter properties, the application of a pre-resolved model ensures that model fitting is not biased by Fe(II) addition.

**Proposed revision to section 3.3:**
The established 13-component PARAFAC model of Cory & McKnight (2005) was used to fit the EEM spectra within this study. The 13 components consist of seven quinone-like fluorophores, including three oxidized quinones (Q1, Q2, and Q3), four reduced quinones (SQ1, SQ2, SQ3, and HQ), two amino acid-like components (tryptophan and tyrosine), and four remaining unknown fluorophores (Cory and McKnight, 2005). We utilized this pre-resolved model as the intent was not to characterize the underlying DOM characteristics at the site, but to examine how the fluorescence characteristics shift with Fe(II), which is known to quench fluorescence. To ensure that this 13-component model adequately represented the fluorescent organic matter characteristics within the sample set, plots of residual fluorescence remaining after the model was applied were plotted and analysed.

**Results**
Line 149 *omit; the lab observation on 1 sample is not a general observation for an entire site*
Agreed. Removed "at the Kidd 2 site" and replaced with "in the groundwater stock solution from the Kidd 2 site."

Line 154 *but the concentration ranges are incomparable; say so*
Agreed.

**Original:** The magnitude of quenching effect was more pronounced in this study than that performed by Poulin et al. (2014), who observed nonlinear fluorescence quenching (7% to 23%) in four different surface water samples, by addition of Fe(II) up to 1.5 mg/L.

**Proposed revision:**
The magnitude of quenching effect was more pronounced in this study than that performed by Poulin et al. (2014), who observed nonlinear fluorescence quenching (7% to 23%) in four different surface water samples, by addition of Fe(II) up to 1.5 mg/L, significantly lower than the Fe(II) concentrations used in this study.

Line 160 *the brings up the question whether the sample cuvetes was covered with a lid during analysis, i.e., could O2 enter freely or only during sample transfer.*

*Note that entrance of $O_2$ may be significant for low Fe(II) but negligible for high Fe(III). Something should be said about this. This might explain the non-linear behaviour or a part of it.*

We agree that the oxidation of Fe(II) during analysis is an important question, but feel the original manuscript addressed the question well. As we have stated in the original text, the analysis time was rapid, we did not observe colloids, and previous work by Poulin suggested that oxidation of Fe(II) to Fe(III) only had a limited effect of fluorescence. We have not made any changes to the text.

*Line 175 seems ugly English to me, although I am non-native English speaker*

Agreed – awkwardly worded. Changed to:

Similar to relative OFI, the relative intensity of Peak A decreased by ~60% as Fe(II) increased from 1 to 306 mg/L, and over 65% of the quenching occurred below Fe(II) concentrations of 101 mg/L (Figure 4c).

*Line 180 this doesn't tell the reader anything. Explain better.*
Agree, we will reword.

Original text:
This result is consistent with quenching experiments conducted with Everglades F1 water samples, where they observed a distinct shift in the quenching locations with increasing ratio of Fe(II) to DOM (Poulin et al., 2014).

**Proposed revision:**
Our results are consistent with the study of Poulin et al. (2014) who observed a shift in the quenching location as the Fe(II) to DOM ratio was increased.

*Line 180 who is "they"*
Agree, this is unclear. We will change "they" to "Poulin et al. (2014)."

*Line 194 compare with remark at Fig. 4. Change y-axis accordingly*

This line states that "High values of FI (approximately 1.80) indicate that DOM is derived from extracellular microbial activity, whereas low values of FI (approximately 1.20) suggest that DOM comes from terrestrial plant and soil organic matter (Cory & McKnight, 2005)."

The y-axis range for FI on Fig 4b was 1.6 to 1.9 and we will revise the y-axis range to be 1.0 to 2.0. We will increase the y-axis ranges for the other parameters on Fig 4 and 5 as recommended by both reviewers.

*Line 203: once more; these studies cannot be intercompared as the Fe range is so different*

We will retain the text as written; Poulin et al. (2014) is the only paper we are aware of that examines this issue, so comparison to their results is reasonable. We acknowledge at several points in the manuscript that their experiments covered a much lower Fe range, and that that is a primary motivation for our study.

**Line 211** *the figure does not show two stages (steps) but two different slopes*
We have changed "occurred in two stages" to "occurred in two phases with increasing Fe(II), as represented by a change in slope."

**Line 215** *emission intensity ??*
Agreed. Changed "intensity at emission" to "emission intensity."

**Line 223** *it is completely unclear how the result of the fitting exercise looks. This is unacceptable.*
As discussed above (see our response to your comment on line 141 and your major comment 3), the data was fit to pre-solved components; we did not generate the components ourselves. We will add plots of the residuals into the supporting information. See our response to your comment on line 141.

We propose to add the following text in this section:
The excitation and emission spectra of the 13 components are included in our dataset published on Zenodo and the residuals from the PARAFAC modeling (measured data minus model fits) are plotted in the supporting information.

**Line 238** *see earlier remark; this finding is not valid for the site as a whole*
Agreed. The wording was not precise. Changed "at the Kidd 2 site" to "in the experimental stock solution from the Kidd 2 site."

**Line 244** *see remark at the figure;*
From Figure 5: *change range of the y-axis. The changes are minor as phrased in the text whereas the figure suggests major changes due to the short range in values.*
We will change the y-axis range for figure 5b to be 0 to 1.

**Line 251** *first word: Iron(II)*
Agreed – changed to "Ferrous iron"

**Line 255** *without reading Senesi (1990), I doubt this. DOC is only 10.7 mg/L whereas Fe goes up to 300 mg/L. This implies less than 1 mmol organic C/l versus almost 6 mmol Fe(II). There thus seems an excess of Fe to me. Comment on this into terms of molar ratios and complexation constants as found for DOM-Fe complexes.*

We agree and have elaborated our discussion.

**Original:**
The non-linear quenching of the fluorescence intensity with Fe(II) concentrations indicates a static quenching mechanism, where quenching primarily depends on the fraction of DOM ligands that are complexed to Fe(II), rather than on the Fe(II) concentration itself (Senesi, 1990).

**Proposed Revision:**
The non-linear quenching of the fluorescence intensity with Fe(II) suggests, following Senesi (1990), a static quenching mechanism. However, our experiment does not allow us to identify quenching mechanisms. While the maximum ratio of Fe(II) to DOM (mg/L per mg/L) was approximately 0.4 in Poulin et al (2014), it is much larger in our study, with a molar Fe to $C_{DOM}$ ratio of approximately 6. While earlier work by Senesi (1990) suggests that quenching primarily depends upon the fraction of DOM ligands that are complexed to Fe(II), our study, with a great excess of Fe(II) over C in DOM, could involve other mechanisms.

Line 259 *see before; do not generalise your findings that much*
Line 260 *explain (comment on mixing of saline and groundwater).*
Line 263 *but Fe is also much higher*

We agree. The effect of mixing was only speculation, so we have removed this from the text.

**Original:**
The differences in Fe(II) quenching at the Kidd 2 site compared to Poulin's study could be related to the high Fe(II) background in the organic-rich environment and the mixing of saline water and groundwater. It should be noted that the DOC concentration in this study was 10.7 mg/L, significantly greater than that used in previous study (Poulin et al. 2014).

**Proposed Revision:**
The differences in the impact of Fe(II) on fluorescence quenching between the sample collected from the Kidd 2 site and the results reported by Poulin et al. (2014) could be related to the much higher Fe(II) background in the organic-rich aquifer and groundwater at the Kidd 2 site. It should be noted that the DOC and concentration in the sample collected from the Kidd 2 site was 10.7 mg/L, significantly greater than that in the previous study (Poulin et al., 2014).

Line 266 *once more*
We have changed "at the Kidd 2 site" to "in the stock solution collected from the Kidd 2 site".

Line 279 *indeed; compare with earlier remark*
Agreed, see response to comment from line 263.

Line 297 *this does not fit to a HESS readership. Explain more*
We believe that DOM composition is relevant to our results and it is useful to suggest this to the reader. Water soluble organic matter is also an important component of continental waters and within the scope of HESS (which includes 'the role of physical, chemical, and biological processes in the cycling of continental water in all its phases, including dissolved and particulate

matter, at all scales'). We have provided more background material in the introduction. No changes made in this section.

**Conclusions**

Line 303 *indicate where the site is as many readers only read CONCLUSIONS*
Agreed. Changed "from the Kidd 2 site" to "from the Kidd 2 aquifer in the Fraser River Delta, Richmond, BC"

Line 304 *should be "concentration is"*
Agreed. Changed to "the DOM concentration is ~10 mg/L."

Line 305  *This point was not addressed before and no new findings should be presented under CONCLUSIONS. Besides, some ideas on Fe-DOC complexation for the different compounds may be feasible. Note that this text piece is somewhat in conflict with the last two sentences of the CONCLUSIONS. Rewrite.*

We agree with the referee that we did not support this statement in the text. We will therefore add the following at the end of the discussion:

[revised manuscript text omitted]

---

## Author Comment (AC3) · 6 Oct 2020

**Response to Referee 2 comment on "Technical note: Effects of iron(II) on fluorescence properties of dissolved organic matter at circumneutral pH"**

Please note the reviewer's comments are in *purple italic Verdana font* and our responses are in Times New Roman, with proposed revisions to the text in blue.

*Anonymous Referee #2*

*This study presents results on the influence of Fe(II) on the fluorescence properties of DOM. The concept is novel and addresses a relevant scientific question. However, the results fail to support the conclusions, specifically the PARAFAC modeling results. My recommendation is that the paper should be reconsidered after major revisions. I have three major comments and the rest are addressed in the comments of the attachment.*

*1. Papers with PARAFAC modeling results typically publish the modeled components and sometimes also the modeled excitation and emission spectra, or at least include them for closer inspection in supplementary material. I'm not sure familiar HESS audience is with PARAFAC, but seeing the modeled components help with understanding.*

*There was also no mention made of your validation techniques. Validating a model can be very subjective. What steps did you take to resolve this dataset into 13 components? I found it difficult to comment on your PARAFAC results without first knowing how you arrived at your modeled results.*

We agree that modeled excitation and emission spectra should be available to support the findings. Indeed, we published these data to a data repository (Zenodo) and referenced the dataset in the "Data availability" section of our paper (http://doi.org/10.5281/zenodo.3737108, please see components_excitation.txt, components_emission.txt, and components_description.txt). HESS policy requests that data should be archived in a FAIR-aligned data repository, not in the supplementary material. In the revised manuscript, we will add a citation to the repository to the references list.

We wish to clarify that we applied a "pre-solved" model, the 13-component model from Cory and McKnight (2005), rather than creating our own model components based on the dataset from Kidd 2. We agree that derivation of a unique PARAFAC model typically requires a large sample set composed of samples from a common organic matter context. The purpose of the manuscript is not to characterize DOM at the site. Rather, it is to understand how ferrous iron affects the fluorescence properties of DOM in groundwater, in which ferrous iron concentrations can be up to 300 mg/L. To ensure that the PARAFAC model is based on a large and diverse sample set and not biased by the presence of ferrous iron, we applied the 13-component model from Cory and McKnight (2005).

We plan to add the following text to the introduction.

**Proposed revision:**
This study fit EEM spectra to a previously derived 13-component PARAFAC model (Cory & McKnight, 2005). We chose to use this robust and well-developed PARAFAC model, which was developed using DOM from a wide range of aquatic environments and has been subsequently applied to interpret EEMs from a large variety of aquatic systems (Jaffé et al., 2008; Larsen et al., 2010). The use of this 13-component model also facilitates the derivation of the redox index (RI), calculated by summing the reduced quinine-like inputs over total quinone-like inputs from components within the model. Finally, derivation of a unique, site-specific PARAFAC model typically requires a large sample set composed of samples from a common organic matter context (Cory & McKnight, 2005; Ishii & Boyer, 2012). As the aim of this study was to capture how spectral attributes are quenched upon addition of Fe(II), rather than characterization of the underlying organic matter properties, the application of a pre-resolved model ensures that model fitting is not biased by Fe(II) addition.

A residual analysis was conducted to assess whether the Cory and McKnight (2005) model was suitable to reproduce the results in this study and we found that there were no significant nor systematic patterns in the residuals. We acknowledge that it was not clear in the original manuscript that we used "pre-solved" components, as Reviewer 1 also had similar misunderstandings regarding the origin of the PARAFAC components used in this study.

*2. I strongly recommend defining optical measurements (absorbance and fluorescence), then describe why they are useful to study DOM, and then define the specific optical properties (FI, HIX, etc.) that were used diagnostically to describe DOM and Fe interactions in this study.*

We agree. Based on feedback from both reviewers, we have expanded the introduction to provide more background and definitions related to absorbance and fluorescence spectroscopy and their use in studying DOM. The end of the introduction now mentions the additional operical properties that were measured, but we leave their definitions for later in the manuscript where the results are reported.

**Proposed revision:**
Moreover, commonly-used fluorescence indices, including fluorescence index (FI) (Cory & McKnight, 2005), humification index (HIX) (Ohno, 2002; Parlanti et al., 2000), the redox index (RI) (Miller et al., 2006), and freshness index (FrI, $\beta/\alpha$) (Parlanti et al., 2000; Zsolnay et al., 1999) were also applied to provide further DOM characterization (section 4.1.3).

*3. This paper could be vastly improved by a more complete and thorough literature review. Some statements were not attributed appropriately/completely. See comments within doc.*

We will update the citations throughout the manuscript and revise the references that were out of date or inappropriately cited. The revised manuscript has ~50% more citations than the original version. A partial list of new references we plan to cite is included in the references list for this document (Aiken, 2014; Bahram et al., 2006; Baker & Spencer, 2004; Bro, 1997; Coble et al.,

2014; Hansen et al., 2018; Helms et al., 2008; R. Jaffé et al., 2008; Rudolf Jaffé et al., 2014; Larsen et al., 2010; Murphy, 2011; Murphy et al., 2013; Nieke et al., 1997; Shen et al., 2020; Stedmon et al., 2003; Stedmon & Bro, 2008; Weishaar et al., 2003; Zepp et al., 2004).

*Comments within PDF:*
*Comment 1 (line 10): Please include a statement about why this study was important from a hydrologic perspective.*

Agreed.

**Original:** The effect of soluble reduced iron, Fe(II), on EEM spectra can be significant, but is difficult to quantitatively assign.

**Proposed revision:** The effect of soluble reduced iron, Fe(II), on EEM spectra can be significant, but is difficult to quantitatively assign, despite the prevalence of groundwater containing high levels of DOM and Fe(II) in deltaic sediments as well as sites contaminated with organics.

*Comment 2 (line 20): This seems like a lot of components for samples that contain NOM.*

We used a pre-resolved model, the Cory and McKnight (2005) 13-component model to characterize the impacts of dissolved Fe(II) on fluorescence properties of DOM. See proposed revisions in response to your major comment 1 above.

*Comment 3 (line 28): Before launching directly into a discussion about fluorescence it is important to first discuss absorbance. I think you would do a great service to your audience to slow down and first define optical measurements (absorbance and fluorescence), then describe why they are useful to study DOM, and finally define the specific optical properties (FI, HIX, etc.) that were used diagnostically to describe DOM:Fe interactions in this study.*

Thank you for this suggestion. We have completely rewritten the Introduction section in response to both referees' feedback (e.g., see our response to Referee 1 major comment 3, and to your major comment 1). We now include background information on the optical measurements and the fluorescence index in this section.

We have retained the explicit definitions of the HIX, RI, etc. in section 4.1.3, close to where the data is reported. Since several indices are presented we think it is useful to report their exact definitions close to where the results are discussed.

For example the following text is retained in section 4.1.3:
"FI is defined as the ratio of emission measured at 470 nm and 520 nm at excitation of 370 nm for instrument-corrected spectra (Cory & McKnight, 2005). High values of FI (approximately

1.80) indicate that DOM is derived from extracellular microbial activity, whereas low values of FI (approximately 1.20) suggest that DOM comes from terrestrial plant and soil organic matter (Cory & McKnight, 2005). Measured FI values increased from an initial 1.62 to 1.80 ($\Delta$FI = +0.18 FI units) with increased Fe(II) concentrations (Figure 4b), indicating the susceptibility of FI to the iron-quenching effect."

*Comment 4 (line 30): This is a review paper...citation is ok, but this statement should be attributed to some others who have advanced work in this field Cory, McKnight, Coble, Aiken, Baker there are so many others!!!*

We agree with the reviewer's literature review assessment. Please see our response to your general comment 3, which details the many of the citations we plan to add in the revised manuscript.

*Comment 5 (line 31): Again so many others to cite here, not just Senesi...Helms, Weishaar for example.*

We have rewritten this section and added the references you mentioned (Helms et al., 2008; Weishaar et al., 2003), among others.

*Comment 6 (line 35): For clarity: In this study we incrementally added Fe(II) to assess the quenching effect on DOM fluorescence in a groundwater sample from a...*

Thank you for the suggestion. The manuscript will be revised as follows:

**Proposed revision:** We incrementally added up to 300 mg/L Fe(II) to DOM-containing groundwater to assess the influence of Fe(II) on the fluorescence properties of DOM. For our analysis, we used groundwater in contact with deltaic sediments in Richmond, British Columbia, Canada, that is representative of other deltaic aquifers, where Fe(II) concentrations can reach >100 mg/L.

*Comment 7 (line 45): Choppy. Should be rewritten more within the spirit of how your work builds upon the work in Poulin et al 2014.*

Agreed.

Original: Similarly, Fe(II) may also interact with DOM to form organometal complexes that could interfere with fluorescence measurements (Poulin et al., 2014). Limited previous research has addressed the quenching effect of Fe(II) interference. Poulin et al. (2014) first demonstrated that fluorescence intensity decreased due to Fe(II)-DOM interactions. Nevertheless, the iron titration experiments were only designed to characterize the Fe(II) quenching effect for surface water with moderately elevated DOM concentrations (2.3 to 5.0 mg/L) under low Fe (II) concentrations (0-1.5 mg/L).

**Proposed revision:** However, limited research has focused on the quenching effect of Fe(II) interference in anoxic groundwater, where reducing conditions are present. Poulin et al. (2014) first demonstrated that Fe(II) and DOM can form organometal complexes that decrease fluorescence intensity. Their experiments were only designed to characterize the Fe(II) quenching effect for surface water with moderately elevated DOM concentrations (2.3 to 5.0 mg/L) under low Fe (II) concentrations (0-1.5 mg/L). To our knowledge, the extent of fluorescence quenching in groundwater with higher Fe(II) concentrations is not known.

*Comment 8 and 9: Are there more citations you could include here to demonstrate a better need for your particular study? Where else are Fe(II) concentrations this high in groundwater? ...see previous comment. What kind of contamination? Are there other studies you can cite?*

Agreed, we have added more citations and context here.

**Proposed revision:** The fluorescence quenching effect in Fe(II)-rich groundwater is still poorly understood and warrants further investigation, given the prevalence of high DOM and Fe(II) in groundwater in deltaic sediments (Bolton & Beckie, 2011) and sites contaminated with organics, for example, from landfills or fuel spills (van Breukelen & Griffioen, 2004; Christensen et al., 2001; Heron et al., 1994). In most instances, these high Fe(II) groundwaters are found when the oxidation of organic matter is coupled to solid-phase Fe(III) reduction, dissolving Fe(II) into groundwater at circumneutral pH. For example, 1.5-10 mg/L Fe(II) in groundwater is commonly observed in the organic-rich groundwaters of the Bengal Basin (Harvey et al., 2002), and up to 90 mg/L Fe(II) has been observed in landfill leachate in the Netherlands (van Breukelen & Griffioen, 2004).

*Comment 10 (line 60): Abstract reports Fluorescence Index, Freshness Index, and HIX. Here you mention Fluorescence Index and Redox Index. Make sure you are consistent.*

Thank you for pointing this out, the manuscript has been revised.

**Proposed revision:** We identified the degree of quenching by Fe(II) based on the excitation−emission matrix (EEMs) regions and peaks, as evaluated using a 13-component parallel factor analysis (PARAFAC) model and commonly-used fluorescence indices, including fluorescence index (FI) (Cory & McKnight, 2005), humification index (HIX) (Ohno, 2002; Parlanti et al., 2000), the redox index (RI) (Miller et al., 2006), and freshness index (β/α) (Parlanti et al., 2000; Zsolnay et al., 1999).

*Comment 11 (line :65) Single space after a period throughout.*

We will use a single space after a period throughout the revised manuscript.

*Comment 12: Just one site and one sample is used as your "stock"? Please explain why this was a representative sample for Kidd 2.*

The purpose of the manuscript is not to characterize DOM at the site where we collected this groundwater sample, but rather to understand how ferrous iron affects the fluorescence properties of DOM in groundwater. We selected our single stock solution because it was typical of the deltaic – aquifer groundwaters that can contain high ferrous-iron concentrations. We have revised the manuscript to make this objective clearer. The "stock" sample was collected at the particular groundwater monitoring well where it had lowest ferrous iron concentration (1.3 mg/L) across the site and therefore most suitable for Fe(II) addition experiment.

*Comment 13 (line 80): Just one 1L bottle was collected? Surely more was collected to conduct your Fe concentration experiments? If I'm confused, so too will be your audience. Please clarify your methods.*

Yes, just one bottle was collected.

**Proposed revision:** For the measurements in this study, we used a single stock solution of natural DOM-containing groundwater, to which we added (titrated) increasing concentrations of Fe(II).

We also quote our response to Referee 1, major comment 1: "The purpose of the manuscript is not to characterize DOM at the site from which the sample (stock solution) was collected. Rather, it is to understand how ferrous iron affects the fluorescence properties of DOM in groundwater. We selected our single stock solution because it was typical of the deltaic aquifer groundwaters that can contain high ferrous iron concentrations."

*Comment 14 (line 88): DOM after filtration will continue to degrade. Are you worried about this? 30 days seems like a lot of time. Many studies of optical properties of DOM show degradation occur in a matter of days, which is why holding time for optical analyses is usually within 2 days of sample collection/filtration. Please provide a justification for this.*

Please also see our response to Referee 1, line 86, who had similar questions.

**Original**
Groundwater was filtered through 0.45 μm cellulose filters, then stored in a 1 L amber glass bottle- with a Teflon-lined cap, without acidification. The bottle was filled with no headspace and duct tape was used to further seal the sample and minimize the oxidation of Fe(II). The collected sample was refrigerated at 4°C until analysis (within 30 days).

**Proposed revision**
The groundwater was filtered through 0.45 μm cellulose filters, then stored in a 1 L amber glass bottle with a Teflon-lined plastic cap, without acidification. The bottle was filled with no headspace and duct tape was used to further seal the sample and minimize the oxidation of Fe(II). The collected 1 L stock solution was refrigerated at 4°C until fluorescence analysis (within ~14 days). Although some degradation of the DOM may have occurred during the holding period, we expect that this would not significantly affect our conclusions as our intention

was to determine how Fe(II) addition affects the fluorescence properties of DOM, rather than to characterize the properties of DOM at the Kidd 2 site.

*Comment 15: (line 115) This sentence should be omitted. The analysis and modeling are two separate things….*

We agree.

**Proposed revision**: The procedures of measuring absorbance and fluorescence of DOM was described by (Hansen et al, 2018). The PARAFAC model was applied to decompose the fluorescence EEMs into chemical meaningful components and provide quantification of the DOM fluorescence spectra (Stedmon, C.A., and Bro, R., 2008). In this study, EEMs were analyzed using previously established 13-component PARAFAC models (Cory & McKnight, 2005).

*Comment 16 (line 125): water Raman. Please cite Murphy 2011 A Note on Determining the Extent of the Water Raman Peak in Fluorescence Spectroscopy*

Thank you, we will cite this reference in the revised manuscript (Murphy, 2011).

*Comment 17 (line 125): I don't know what this means. Do you mean the spectra were normalized to the daily water Raman, hence fluorescence intensity units are reported as RU...?*

Thanks for pointing this out. The fluorescence EEMs were generated, not intensity. The manuscript will be revised.

**Original:** Subsequently, the fluorescence intensity (presented in Raman units (RU)) was generated as a function of the excitation and emission wavelengths.

**Proposed revision:**
Fluorescence intensity within all EEM data is presented in Raman units (RU) due to the way that raw EEM spectra are corrected prior to analysis via PARAFAC modelling or calculation of associated indices. As per standard practice, raw EEMs were instrument corrected via software provided by the instrument manufacturer. Spectra were then corrected for inner filter effects (Ohno, 2002), then normalized to the area under the Raman curve (Nieke et al., 1997; Stedmon et al., 2003); second order Raleigh scatter and Raman bands were excised at a bandpass of 12 nm (Bahram et al., 2006; Zepp et al., 2004) while first order Raleigh scatter was excised at a bandwidth of 50 nm to remove all spectral artefacts (Bro, 1997; Stedmon & Bro, 2008). Specifically, normalization to the area under the Raman curve (which occurs due to the inelastic scatter of light by water) contributes to instrument correction that allows for the comparison of spectra between different instruments and thus different studies.

*Comment 18 (line 135): How many spectra were included in your PARAFAC dataset? I would hesitate to try to validate a model containing just a few samples. Typically PARAFAC is best applied to large datasets (n>100). I would like to see the modeled excitation and emission spectra. Additionally, please also provide the modeled components. Modeled components are usually published in a paper that contains this method. Also, seeing the components will be especially helpful to an audience not familiar with the model output. Also not included in your text here is the model validation results. Validating a model can be very subjective. What steps did you take to resolve this dataset into 13 components? Please read Murphy et al 2013 for a detailed walthrough of PARAFAC model validation techniques. Fluorescence spectroscopy and multi-way techniques. PARAFAC. I have to say that 13 components seems like a lot for groundwater.*

Please see our detailed response to your general comment 1. We used pre-solved components from Cory and McKnight (2005) and will update the manscript to clarify this. Our dataset published on Zenodo already includes the excitation and emission spectra for all 13 components. As recommended in the author guidelines for HESS, we published the data in a FAIR-compliant data repository rather than as supplementary materials on the journal website.

*Comment 19 (line 149): Refer here to the sample name, not the site. One location is not indicative of DOM at the entire site.*

Please refer to our response to your comment # 12. The stock water represents deltaic aquifer groundwater at Kidd2 site, where groundwater that can contain high ferrous iron concentrations.

*Comment 20 (line 154): Careful the study by Poulin et al 2014 used samples where concentration ranges were not comparable to those used here.*

**Original:** The magnitude of quenching effect was more pronounced in this study than that performed by Poulin et al. (2014), who observed nonlinear fluorescence quenching (7% to 23%) in four different surface water samples, by addition of Fe(II) up to 1.5 mg/L

**Proposed revision:**
The magnitude of quenching effect was more pronounced in this study than that performed by Poulin et al. (2014), who observed nonlinear fluorescence quenching (7% to 23%) in four different surface water samples, by addition of Fe(II) up to 1.5 mg/L, significantly lower than the Fe(II) concentrations used in this study.

*Comment 21 (line 175): Please clarify this sentence.*
**Original:**
Similar to relative OFI, the relative intensity of Peak A decreased by ~60% as Fe(II) increased from 1 to 306 mg/L, and over 65% of the quenching was occurring when Fe(II) reached to 101 mg/L (Figure 4c).

**Proposed revision:**
The relative intensity of Peak A decreased approximately 60% as Fe(II) increased from 1 to 306 mg/L. Of the total decreased intensity, over 65% of the quenching occurred when Fe(II) increased from 1 to 101 mg/L (Figure 4c).

*Comment 22 (line 176): For clarity I would say: shorter (higher energy) emission wavelengths.*

**Original:** In addition, the position of Peak A continuously migrated toward the shorter emission wavelengths with a constant excitation wavelength of 239 nm and increasing Fe(II) concentration.

**Proposed revision:** In addition, the position of Peak A continuously migrated toward the shorter (i.e., higher energy) emission wavelengths with a constant excitation wavelength of 239 nm and increasing Fe(II) concentration.

*Comment 23 (line 178): Again for clarity because the wavelengths didn't change, the location of the fluorescence response changed: ...overall the emission of the fluorescence response shifted from 441 nm to 409 nm as Fe(II) increased...*

**Original:** Although the linear relationship was not observed, overall the emission wavelength gradually changed from 441 to 409 nm as Fe(II) increased from 1 to 306 mg/L.

**Proposed revision:** Although the linear relationship was not observed, overall the location of fluorescence response gradually changed from 441 to 409 nm as Fe(II) increased from 1 to 306 mg/L.

*Comment 24 (line 191): When defining each of these optical properties, it would be useful to state an expected (typical) range of concentrations from indices used in other groundwater DOM studies. There will be lots of studies to cite here and will provide readers with an idea of where results from this study fall within the literature as a whole.*

We agree. The typical range for fluorescence index (FI) and redox index (RI) were already reported (line 194 and 243). We will add ranges for HIX and FrI.

[revised manuscript text omitted]

---

## Author Response (AR1)

Dear Professor Vanclooster:

We are pleased to submit our revised manuscript "Technical note: Effects of iron(II) on fluorescence properties of dissolved organic matter at circumneutral pH" as well as a tracked changes version of the document. We are grateful for the referees' constructive comments, which helped us to improve the manuscript. Our detailed point-by-point responses to the referees are available online (see: https://hess.copernicus.org/preprints/hess-2020-150/). We summarize the changes to the manuscript below.

We have clarified in the revised manuscript that the established 13-component PARAFAC model of Cory & McKnight (2005) was used to fit the excitation-emission matrices (EEMs) in this study. We acknowledge the original manuscript was not sufficiently clear on the origin of the PARAFAC model because both referees had misapprehensions about this aspect of our methodology. We chose to use an existing model based on a large and diverse dataset so we could characterize how spectral attributes were quenched by the addition of ferrous iron.

We have rewritten and expanded the introduction to make the manuscript more accessible to the broad HESS audience, providing more background on fluorescence spectroscopy methodology and the motivation for this study in the context of prior work. We have added new citations throughout the manuscript to better reflect the existing literature (the total number of references has increased by ~50%). We have also made numerous minor changes as requested by both referees, for example, modifying the axis ranges on Figures 4 and making wording changes to clarify imprecise statements and correct grammar.

Thank you for considering our revised manuscript.

Sincerely,

Cara C. Manning on behalf of coauthors

[revised manuscript text omitted]

---

## Author Response (AR2)

Dear Professor Vanclooster:

We are pleased to submit our newly revised manuscript "Technical note: Effects of iron(II) on fluorescence properties of dissolved organic matter at circumneutral pH" as well as a tracked changes version of the document. We are grateful for the constructive comments of Referee 3 and are thankful to them for agreeing to review the revised manuscript and assess the changes that we made in response to Referee 1 and Referee 2's comments. Our detailed point-by-point responses to their comments are available below. The Referee's comments are in black Arial font, our responses are in purple Times New Roman, and our revised text is in blue Times New Roman.

Thank you for considering our revised manuscript.

Sincerely,

*Cara Manning*

Cara C. M. Manning on behalf of coauthors

Comments from Referee 3:
Overall, the revisions performed by the authors are acknowledged. However, some issues still require further clarification:

1. The effect of the added SO4 ions is still not accounted for. Could SO4 ions in the added concentration affect the obtained fluorescence spectra and indices? Is the spiked solution comparable to the naturally-occurring high-Fe(II) concentration water in the deltaic aquifer? This needs to be addressed (either by referencing previous research or experimentally).

Thanks for this suggestion to clarify the potential effects of $SO_4^{2-}$ addition via the $FeSO_4(H_2O)_7$ spike and compare the ion concentrations in the spiked solution to the natural conditions in the aquifer. We note that Poulin (2014) also prepared iron stock solutions with $FeSO_4(H_2O)_7$ and reported a significant quenching effect caused by the binding of Fe(II). The stock solution was taken from the deep layer of the aquifer, where we previously measured the ambient concentration of $SO_4^{2-}$ was 71 mg/L (Jia, 2015). Therefore, as we increased Fe(II) from 1.3 to 306 mg/L (a factor of 240 increase), the $SO_4^{2-}$ likely increased from approximately 71 to 595 mg/L (a factor of 8 increase). In our revised text below, we demonstrate that the Fe, $SO_4^{2-}$ and Cl concentrations in the experimental spiked solution are similar to the natural conditions occurring in the aquifer. We therefore believe that the dominant effect we observed was the effect of addition of Fe(II).

Revised section 3.2 (new text in **bold**):
"An Fe spiking solution of 1000 mg/L Fe(II) was prepared with $FeSO_4(H_2O)_7$, **following Poulin et al. (2014)**, using the DOM stock solution so that spiking with Fe(II) would not

change the overall concentration of DOM. … **Previous analyses of water from W3-14 via ICP-OES (Jia, 2015) indicated the $SO_4^{2-}$ concentration was 71 mg/L and $Cl^-$ concentration was 1670 mg/L. Therefore, as the Fe(II) concentration was increased by a factor of 240 (from 1.3 to 306 mg/L), the $SO_4^{2-}$ concentration only increased by a factor of 8 (from approximately 71 to 595 mg/L). We therefore expect that the dominant effect observed through this addition experiment is the effect of increasing Fe(II), rather than the effect of increasing $SO_4^{2-}$ and/or total anions. The anion and cation concentrations in the experimental spiked solution were similar to the natural conditions occurring in the aquifer. For example, for the depths with $Fe^{2+}$ from 50–435 mg/L, the range in $SO_4^{2-}$ was 13–600 mg/L and range in $Cl^-$ was 50–9600 mg/L (Jia, 2015)."**

2. Since the water sample used in this work was artificially spiked with an Fe(II) salt, a comparison between the highest concentration sample and the naturally-occurring high-Fe(II) concentration water in the deltaic aquifer would significantly strengthen the paper.

Agreed. As mentioned in the response to point 1 above, we have added a comparison between the natural and spiked waters to the manuscript. The thesis of Jia (2015) presents a full geochemical characterization of the Kidd II aquifer, including EEM spectra of the natural water samples (thesis figure 4.36). We plan to submit a site characterization paper separately, and decided it would be best to first publish a technical note describing and validating our approach of using EEM spectra in waters with high Fe(II), which we can then reference in the site characterization paper.

3. In the description of the FI (lines 239-243) the authors clearly state that the value of the index indicates the source of the organic matter. The following paragraph describes a change in FI caused by the increase in Fe(II) concentrations. However, the addition of Fe(II) clearly did not change the source of the organic matter in the system. This can be confusing to the reader and thus should be clarified and properly explained.

Thanks, we have clarified this description.

Changed to (new text in **bold**):
"**In the absence of fluorescence quenching by other dissolved constituents**, high values of FI (approximately 1.80) indicate that DOM is derived from extracellular microbial activity, whereas low values of FI (approximately 1.20) suggest that DOM comes from terrestrial plant and soil organic matter (Cory & McKnight, 2005)."

4. The last section of the introduction (lines 79-92) is very long and contains a description that is more suitable for the 'Methodology' section. I recommend shortening this section significantly.

We have shortened this section; it is now three lines long instead of 13. We realized that the text is repetitive to points already in the methodology, section 3.3 (line 177-189), so we eliminated some text in the introduction and referenced section 3.3.

Revised text in introduction (new text in **bold**):

"We identified the degree of quenching by Fe(II) based on the excitation-emission matrix (EEMs) regions and peaks. In this study we fit EEM spectra to a previously derived 13-component PARAFAC model (Cory & McKnight, 2005); **see section 3.3 for further details**."

5. In line 281 the authors state that the residuals of the PARAFAC model are plotted in the supporting information. However, I wasn't able to find a supporting information file in your submission which makes the evaluation of this part difficult (I did find the online datasets, but these contain only .txt files and no plotted data).

We apologize for the confusion caused by our inaccurate statement that the residuals of the PARAFAC model are plotted in the supporting information. We had considered adding these plots but ultimately decided against it. We have removed this statement from the revised manuscript.

To provide you with confidence in our approach of applying the Cory & McKnight (2005) PARAFAC model and to demonstrate that the residuals were negligible, we have provided plots of the data, model and residuals below for three of the samples (1, 131, and 306 mg/L $Fe^{2+}$). The values in the colorbar of each figure represent the fluorescence intensity in Raman Units for a given excitation and emission wavelength.

These figures are consistent with our statements on line 181 of the revised manuscript: "To ensure that this 13-component model adequately represented the fluorescent organic matter characteristics within the sample set, the residual fluorescence remaining after the model was applied were plotted and analysed. No systematic residuals were found after fitting the EEMs to the PARAFAC model, suggesting that the model was able to represent the samples, and that Fe(II) additions did not significantly change the structure of fluorophores in the groundwater stock solution from the Kidd 2 site."

We decided against including these figures in a supporting information file because they were a different visualization of the data than the Figure 2 in the main manuscript and we thought that it may cause confusion to show two different visualizations of the EEMs. We generated these figures in 2013 and did not archive the output values for the model and residuals. We found that it was too difficult to exactly reproduce all the steps to generate the residuals and re-plot the data 7 years after the original analysis was performed, and therefore we decided against including plots of the residuals in the supporting information.

We believe that the figures below will convince you that no systematic residuals are present when fitting our sample EEM data to the Cory & McKnight (2005) model.

[Figure]

1.3 mg/L Fe$^{2+}$

131 mg/L Fe$^{2+}$

6. The description of the main experiment and the aquifer location in line 48 appears again in the context of the objective (lines 71-76). This is an unnecessary repetition. I recommend omitting lines 48-50. The text in lines 46-48 can be joint to the following paragraph.

We agree that lines 48-50 and 71-76 are repetitive. As suggested, we removed lines 48-50 ("In this study, an Fe(II) addition experiment was performed to assess the quenching effect of Fe(II) in groundwater samples from a deltaic aquifer in Richmond, British Columbia, Canada, where natural Fe(II) concentrations reach over 300 mg/L (5.4 mM).") and combined lines 46-48 into the following paragraph.

7. The 'Conclusions' section currently fits the description of a summary.

Changed section title from "Conclusions" to "Summary".

8. I find the separation between the results and the discussion unnecessary. However, I leave that to the authors' consideration.

We decided to maintain the separation between results and discussion. The results section is more structured (containing 4 subsections, each on a different ) whereas the discussion goes through the results holistically in a single section

Technical comments:

1. Line 37: consider omitting the word 'Additionally'.

Agreed, omitted.

2. Section 3.3: this section described data analysis rather than sample analysis. I recommend moving the first two lines (148-149) to the previous paragraph that describes the specifics of the Fe(II) addition experiment and changing the title to 'Data analysis'.

Agreed, we have changed the section 3.3 title from "Sample analysis" to "Fluorescence data acquisition and analysis." Lines 148-149 which describe the Fe(II) concentration determination are moved to the previous section 3.2 which we have changed from "Fe(II) addition experiment" to "Fe(II) addition experiment and concentration determination."

3. Line 137: 'An Fe spiking…' instead of 'A Fe spiking…' (same as you correctly wrote in line 49).

Done.

4. Line 154: 'collected at a bandpass of 5 nm' instead of 'collected at a bandpass at 5 nm'.

Done.

5. Line 165: omit 'then'.

Omitted.

6. Line 174: change '=' to 'of'.

Done.

7. Section 4.1: the title of the section is awkwardly phrased. I recommend changing it to either 'The effect of Fe-quenching on EEM fluorescence or at least 'Fe-quenching effect on EEM fluorescence' (same comment stands for section 4.2).

Agreed, we've changed the section 4.1 title to "The effect of Fe(II) quenching on EEM fluorescence" and changed the section 4.2 title to "The effect of Fe(II) quenching on PARAFAC modeling and component distribution"

8. Line 218: add references.

Agreed, changed as described below.

**Original text:** "Coble (1996; 1990) identified five primary peaks from a visual inspection of EEMs, including humic-like Peaks A, C, and M; and protein-like Peaks B and T. The peaks are believed to be linked to the organic matter properties, and have been used for fluorescence comparisons in numerous studies. Coble (1996; 1990) identified five primary peaks from a visual inspection of EEMs, including humic-like Peaks A, C, and M; and protein-like Peaks B and T."

**Revised text:** "Many studies have characterized fluorescence properties of waters based on the primary peaks in EEM spectra, identified by visual inspection and/or multivariate data analysis (Chen et al., 2003; McKnight et al., 2001; Murphy et al., 2013; Shen et al., 2020; Stedmon et al., 2003; Stedmon & Bro, 2008). The positions of these peaks are believed to be linked to the organic matter properties. Coble (1996; 1990) identified five primary peaks from a visual inspection of EEMs, including humic-like Peaks A, C, and M; and protein-like Peaks B and T."

9. Line 226: 'Although a linear relationship…' instead of 'Although the linear relationship…'.

Changed.

10. Line 251 'they' instead of 'he' (the cited paper has multiple authors).

Changed.

11. Line 269: 'Similarly' instead of 'Similar'.

Changed from "Similar to HIX" to "Similar to trends for HIX" on line 269. Analogous correction performed on line 220.

12. Line 278: omit the word 'yet'.

Omitted.

Thank you again for taking the time to provide very helpful feedback on our revised manuscript and to acknowledge the changes we made in response to the other reviewers.

---

## Author Response (AR3)

Dear Professor Vanclooster:

We are pleased to submit our newly revised manuscript "Technical note: Effects of iron(II) on fluorescence properties of dissolved organic matter at circumneutral pH" as well as a tracked changes version of the document. We are grateful to reviewer 3 for taking the time to review our revised mansucript and for their positive feedback.

Our detailed point-by-point responses to their comments are available below. The Referee's comments are in black Arial font, our responses are in purple Times New Roman, and our revised text is in blue Times New Roman. We have also revised the formatting of the references so it matches the Copernicus format.

Thank you for considering our revised manuscript.

Sincerely,

Cara C. M. Manning on behalf of coauthors

The revisions performed by the authors according to my previous comments are acknowledged and I'm generally pleased with the revised version.

Thank you for your positive feedback.

I still think that the following points need further attention before publication:

1. The end of the introduction still needs work. The introduction should preferably end with a single concise paragraph describing the aim of the study and briefly the method. In the latest version there are still two separate paragraphs for this purpose. Additionally, the reference of the methodology section in the introduction seems unnecessary.

Based on this feedback, we combined the last two paragraphs of the introduction into a single paragraph, and revised the wording. We also removed the reference to the methodology section. Revised text:

The objective of this study was to assess the influence of high concentrations of Fe(II) on the fluorescence properties of DOM by titrating up to 306 mg/L (5.4 mM) Fe(II) into groundwater collected from a deltaic aquifer in Richmond, British Columbia, Canada. This groundwater is representative of groundwater found in diagenetically immature, organic-rich deltaic sediments, where Fe(II) concentrations can reach up to 300 mg/L (Bolton & Beckie, 2011; Jia, 2015). The biogeochemistry of groundwater at this site, and an analysis of the origin of the extraordinarily high Fe(II) concentrations, are described in Jia (2015).

In this study, we identified the degree of quenching at different Fe(II) concentrations (from 1 to 306 mg/L) based on the excitation-emission matrix (EEMs) regions and peaks We fit the EEM spectra to a previously derived 13-component PARAFAC model (Cory and McKnight, 2005) and calculated commonly-used fluorescence indices to quantify DOM fluorescence properties as a function of Fe(II) concentration. This study provides a detailed characterization of the impact of changing Fe(II) concentrations on DOM fluorescence.

2. The phrasing you chose for the sentence in line 274 (in the revised version) seems incorrect. I suggest something like: 'Similarly to the trend obtained/observed for HIX...'. Same comment applies to line 226.

Agreed.
Line 274 changed from "Similar to trends for HIX" to "Similar to the trend observed for HIX"
Line 227 changed from "Similar to trends for relative OFI" to "Similar to the trend observed for relative OFI"

3. Paragraphs 2 and 3 of the discussion can be combined.
Agreed. We have combined paragraphs 2 and 3. Revised text:

Poulin et al. (2014) mainly examined the effect of Fe(II) addition to terrestrial-derived fresh surface water with undetectable Fe(II) levels. In contrast, the DOM in the stock solution collected from the Kidd 2 site is hypothesized to be derived from microbial sources and may respond to high Fe(II) concentrations differently than freshwater terrestrial-derived DOM. Quenching of humic-like peaks by other metal species has been observed by other researchers. Ohno et al. (2007) conducted experiments on the impact of Fe(III) and Al(III) addition to the deciduous water-soluble organic matter (WSOM) fluorescence spectra. This result showed that the fluorescence intensity was quenched by about 30% in the presence of 25 μM (1.4 mg/L) Fe for Peak A (Ohno et al., 2007). The DOM fluorescence quenching mechanism by metals is not well understood.